# Mechanistic Behavior Editing of Language Models

## Abstract

Large Language Models trained on web-scale text acquire language generation abilities that can solve a wide range of tasks, particularly when task knowledge is refined into the generative prior using in-context examples. However, spurious features learned from noisy data hinder their generalizability. Supervised finetuning can introduce task specificity, but introduce data inefficiency. Prior studies indicate that (i) noisy neural circuitries coexist with generalizable ones within LLMs, and (ii) finetuning typically enhances (or suppresses) existing abilities without introducing newer ones. Building upon these, we propose `TaRot`, a novel method for task adaptation. `TaRot` intervenes in the neural circuitries using learnable rotation matrices that are optimized using Bayesian Optimization, on labelled samples in the order of standard few-shot prompting examples. Experiments on multiple classification and generation tasks using LLMs of varying sizes reveal the efficacy of `TaRot`, improving upon both zero- as well as few-shot performance, with average improvements (across models and tasks) of 23.81% and 11.15%, respectively.

## 1 Introduction

Large Language Models (LLMs) acquire the ability to associate different language concepts presented in a sequential context by optimizing the prediction probability of the next token given a context. Despite its apparent simplicity, when scaled across web-sized text corpora, such a learning strategy introduces the ability to solve a wide range of tasks presented in natural language. However, the web contains almost everything humankind has written, and therefore, it introduces spurious token associations that are irrelevant or even counter-productive to the model to become generalized task-solvers. We observe phenomena like brittle few-shot performance (Sclar et al., 2024), hallucination (Huang et al., 2023), harmful text generation (Wen et al., 2023), etc. as evidence of learning noisy patterns. Remedial interventions like instruction tuning (Zhang et al., 2024), alignment tuning (Shen et al., 2023), etc. have been proposed. Recent research has shown that such mediation only acts on a superficial level — out-of-distribution inputs can reinforce noisy behavior and break the model (Ghosh et al., 2024). Without an in-depth understanding of the inner workings, remedial strategies become wild goose chase.

Mechanistic disentangling of Transformer-based language models has shed some light on this direction (Elhage et al., 2021; Olsson et al., 2022; Wang et al., 2023). Two recent investigations (Jain et al., 2024; Prakash et al., 2024) on the effects of fine-tuning confirm the inability of supervised fine-tuning to alter fundamental abilities acquired via pretraining. On a tangential investigation, Dutta et al. (2024) recently confirmed the existence of multiple parallel neural pathways of answer processing within LLMs. Bhaskar et al. (2024) echoed similar findings in the case of syntactic generalization while pointing out that different components acquire different generalization behaviors. These findings lead us to the central research question of this work: *is it possible to directly edit the model behavior via mechanistic interventions in a generalizable manner?* Prior work in this direction has heavily relied on careful manual effort to localize task-specific neural components and design intervention techniques Meng et al. (2022); Li et al. (2024a). Two shortcomings hinder the widespread use of such methods: (i) Complexity of localization increases polynomially with model size; identifying which component is responsible for each different task and designing suitable ablation is extremely challenging. (ii) The existence of multiple components performing similar neural computations within the model challenges the generalizability of the intervention itself.

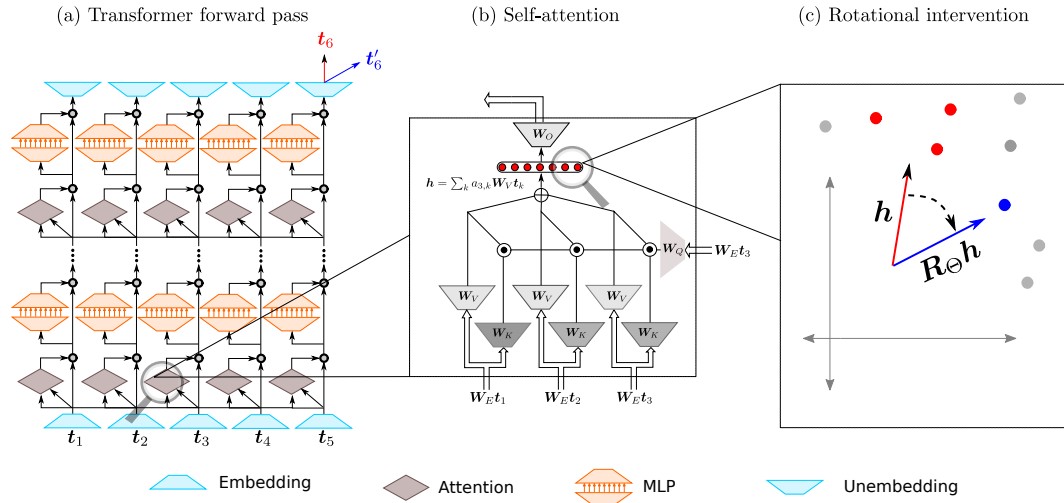

Figure 1: **A conceptual illustration of `TaRot`.** (a) A token sequence $[t_1, t_2, t_3, t_4, t_5]$ is input to a pretrained language model, generates an undesired next token $t_6$. (b) A certain attention head is responsible for associating input tokens $t_1$, $t_2$, $t_3$ with the undesired output via pretrained memorization. These associations are memorized through the OV-circuit of the attention head. (c) The direction of the attention-weighted sum of the value vectors, $h$, is aligned to the undesired token directions (shown in red). `TaRot` learns a parametrized rotation operator $R_\Theta$ that rotates $h$ to the direction of the desired token direction (shown in green). The intervention results in a change in the forward pass in (a) that outputs $t_6'$.

**Our contribution.** To this end, we propose a novel intervention technique, `TaRot` – **T**ask-**a**ware **Rot**ation of token-association (see Figure 1 for a representative depiction)[1]. We establish the conceptual prior from Transformers's implicit gradient descent bias in next token prediction. Specifically, we first show that attention-weighted averaging of value vectors facilitates the memorization of token association from pertaining data in individual attention heads, in the sense that each attention head acts as a mini-language model. Due to the vast number of token associations present in the pretraining corpus compared to the number of attention heads in even the largest of the models, we hypothesize that individual directions of these memorized associations remain in superposition, and removal or downscaling of a head can counteract model performance. Instead, we construct parametrized rotations to align head outputs for task-adaptation. The rotation parameters are then optimized using Bayesian optimization. Furthermore, `TaRot` is *extremely data- and compute-efficient*: we use 6-20 supervised examples for each task and $\frac{dL}{4}$ rotation parameters (where $d$ is the model dimension and $L$ is the number of layers) for each different task. This renders `TaRot` at par with standard few-shot prompting in labeled data-efficiency.

We experiment with five different classification tasks and two natural language generation tasks; the choice of tasks seeks to investigate general world knowledge (news topic classification) as well as the ability to generalize beyond imitation (BIG Bench tasks (BIG-bench authors, 2023)). `TaRot` demonstrates consistent improvements over four different language models of varying sizes: Qwen2-1.5B-Instruct, Phi-3-mini-4k-instruct, Mistral-7B-Instruct-v0.1, and Meta-Llama-3-8B-Instruct, in both zero-shot as well as few-shot settings. Furthermore, we analyze the changes in neural representation introduced by `TaRot` to uncover useful insights.

## 2 RELATED WORK

Our work is primarily relevant to two broad areas of existing literature: adaptation of pretrained language models to downstream tasks, and mechanistic understanding and intervention techniques.

---

[1]The source code of `TaRot` is attached with the supplementary and will be made public upon acceptance of the paper.

**Task adaptation of pretrained language models.** The *pretrain-finetune* regime for adapting language models to downstream tasks dates back to the early approaches like BERT (Devlin et al., 2019) — pretrain a language model (LM) on large unstructured text corpora using self-supervised objective, followed by supervised fine-tuning on task-specific, relatively smaller datasets. Despite the apparent simplicity, the pitfalls of this regime have been pointed out in terms of *distribution shift* (Kumar et al., 2022). With the development of large-scale, autoregressive Transformer-based language models and their ability to learn from in-context examples (Brown et al., 2020), a definitive shift has happened in the more recent past. Current practices of using these models for downstream tasks primarily rely on designing suitable prompt templates and labeled example retrieval for in-context learning (ICL) (Liu et al., 2022; Rubin et al., 2022; Tanwar et al., 2023); traditional techniques of fine-tuning have taken a back seat due to the computational cost and catastrophic forgetting introduced by small-scale task-specific data that hurts the pretrained abilities (Zhai et al., 2024). Instead, finetuning to follow task instructions, *aka* instruction-tuning (Zhang et al., 2024), has gained popularity. Instruction-tuning has been shown to introduce zero-shot task adaptation abilities in LLMs (Wei et al., 2022). Additionally, different methods of alignment tuning have been proposed with the primary goal being aligning the generative distribution of the language models with human values and preferences (Shen et al., 2023; Wang et al., 2024b). Despite the popularity of instruction and alignment tuning, their ability to alter fundamental information processing has been put in question in recent literature. Jain et al. (2024) investigated the effects of fine-tuning in toy models trained with formal languages as well as precompiled ones; their findings suggest that supervised fine-tuning does not introduce any new ability into pretrained models but only reinforces (or suppresses) existing ones. Similar concerns have been raised upon investigating entity tracking in the neural representation space (Prakash et al., 2024). Ghosh et al. (2024) identified multiple limitations of instruction tuning, including the inability to introduce new knowledge and deterioration of performance due to over-reliance on pattern matching.

**Mechanistic understanding and interventions.** The umbrella of mechanistic interpretability broadly encompasses methods to disentangle model behavior via reverse engineering the underlying neural algorithm (Elhage et al., 2021; Ferrando et al., 2024). Endeavors to mechanistically understand Transformer-based language models trace back to the seminal work by Elhage et al. (2021). Their framework established attention heads as one of the fundamental building blocks of language model interpretation. Subsequent studies have identified the functional roles of different attention heads in pretrained models: induction heads as a primary mechanism of prefix matching (Olsson et al., 2022), circuitries of attention heads responsible for indirect object identification (Wang et al., 2023), neural pathways that implement chain-of-thought reasoning (Dutta et al., 2024), etc. Much relevant to our analysis, Lv et al. (2024) found that certain attention heads memorize the association between country names and their capitals. On a tangential line of investigation, Geiger et al. (2024) introduced the Distributed Alignment Search (DAS) framework for localizing interpretable features in subspaces of the neural representations. Mechanistic methods provide actionable insights that have led to non-traditional techniques to edit model behavior. Elhage et al. (2021) experimented with key propagation to elicit induction heads (and thereby, prefix-matching ability) in single-layer attention-only Transformers. Meng et al. (2022) used causal tracing to locate factual associations in MLP neurons and proposed a gradient-free approach to edit factual recall patterns in pretrained language models. Li et al. (2024a) identified attention head circuitry that elicits toxic text generation in GPT-2; mean-ablation of these circuits is shown to reduce toxicity. Self-detoxification (Leong et al., 2023) identifies toxic generation direction in the internal representation using trigger prompts and then rewrites in the opposite direction to reduce toxicity. Wang et al. (2024a) formulated toxicity reduction as a knowledge editing task that can permanently alter toxic behaviors instead of suppressive interventions like supervised fine-tuning or RLHF-based alignment. Lamparth & Reuel (2024) localized backdoor mechanisms (i.e., vulnerabilities against adversarial prompt injections) in early-layer MLPs and proposed a low-rank substitution to improve robustness against such injections. Vergara-Browne et al. (2024) employed attribution patching techniques to identify and remove certain singular values in the parameter matrices to improve performance.

In comparison with prior intervention approaches, our work bears two fundamental differences: (i) `TaRot` does not necessitate task-specific localization of neural behaviors; this significantly reduces intense manual effort and risk of over-localization, eliciting efficient, generalizable interventions; (ii) `TaRot` is gradient-free, parameter-efficient, and requires supervised samples in the order of standard ICL; this poses `TaRot` as a practical alternative to intense prompt-engineering.

## 3 METHODOLOGY

In this section, we demonstrate the role of attention heads in memorizing token associations. Next, we lay out the working principles of `TaRot`.

### 3.1 ATTENTION HEADS AS TOKEN-TOKEN MAPS

Following the framework presented by Elhage et al. (2021), we dissect the Transformer-based language models with the following assumptions: (i) Each attention head reads from and writes to the residual stream independently in a linear fashion, and (ii) given that the attention heads utilize hidden representation of dimensionality much smaller than the residual stream (i.e., for a model with 16 attention heads, each attention head uses 1/16-th of the dimension of the residual stream), they typical operate on small subspaces of the residual stream. This way, two attention heads can operate on two distinct subspaces and never interact with each other. These two assumptions allow us to interpret the working of the attention heads meaningfully even while treating each head in isolation. We start with identifying what a single-head attention operation tends to learn in isolation.

Following the standard terminology (Elhage et al., 2021), we represent the embedding and unembedding matrices as $\boldsymbol{W}_E \in \mathbb{R}^{d \times V}$ and $\boldsymbol{W}_E \in \mathbb{R}^{V \times d}$, where $d$ and $V$ are the dimensionality of the residual stream and the token space, respectively, the query, key, value, and output projection matrices denoted as $\boldsymbol{W}_Q, \boldsymbol{W}_K, \boldsymbol{W}_V, \boldsymbol{W}_O \in \mathbb{R}^{d \times d}$, respectively. Given a sequence of input tokens as one-hot column vectors $\boldsymbol{T} = \{\boldsymbol{t}_1, \cdots, \boldsymbol{t}_n\}$, the forward pass for single-layer attention-only Transformer can be written as:

$$\hat{\boldsymbol{t}}_{n+1} = \boldsymbol{W}_U \left( \mathbf{W}_E \boldsymbol{t}_n + \boldsymbol{W}_O \sum_i a_{n,i} \boldsymbol{W}_V \boldsymbol{W}_E \boldsymbol{t}_i \right) \tag{1}$$

where $a_{n,i} = \frac{\exp\left(\boldsymbol{t}_n^\top \boldsymbol{W}_E^\top \boldsymbol{W}_Q^\top \mathbf{R}_{\Theta, n-i} \boldsymbol{W}_K \boldsymbol{W}_E \boldsymbol{t}_i\right)}{\sum_j \exp\left(\boldsymbol{t}_n^\top \boldsymbol{W}_E^\top \boldsymbol{W}_Q^\top \mathbf{R}_{\Theta, n-j} \boldsymbol{W}_K \boldsymbol{W}_E \boldsymbol{t}_j\right)}$ is the softmax-attention probability from source token $\boldsymbol{t}_i$ to destination token $\boldsymbol{t}_n$, and $\hat{\boldsymbol{t}}_{n+1} \in \mathbb{R}^V$ is the logit of the predicted next token. Upon reparametrization of $\boldsymbol{W}_U \boldsymbol{W}_O \boldsymbol{W}_V \boldsymbol{W}_E$ as $\boldsymbol{W}_{OV}$, we can rewrite Equation 1 as

$$\hat{\boldsymbol{t}}_{n+1} = \boldsymbol{W}_U \mathbf{W}_E \boldsymbol{t}_n + \sum_i \boldsymbol{W}_{OV} \boldsymbol{t}_i \tag{2}$$

Note that $\boldsymbol{W}_{OV} \in \mathbb{R}^{V \times V}$, denoted as OV-circuits by Elhage et al. (2021), maps a distribution over tokens to another distribution over tokens. If the true token is $\boldsymbol{t}_{n+1}$ with $I(\boldsymbol{t}_{n+1})$ donating its index (i.e., index of 1 in $\boldsymbol{t}_{n+1}$), then the typical language modeling loss can be calculated as:

$$\mathcal{L}(\hat{\boldsymbol{t}}_{n+1}, \boldsymbol{t}_{n+1}) = -\log \left( \frac{\exp\left(\hat{\boldsymbol{t}}_{n+1}^{(I(\boldsymbol{t}_{n+1}))}\right)}{\sum_k \exp\left(\hat{\boldsymbol{t}}_{n+1}^{(k)}\right)} \right) \tag{3}$$

We can compute the gradient dynamics of the OV-circuit (with unit batch size and zero momentum) using Equations 2 and 3 as follows:

$$\boldsymbol{W}_{OV}^{(s+1)} = \boldsymbol{W}_{OV}^{(s)} + \eta \boldsymbol{t}_{n+1} \left( \sum_i a_{n,i} \boldsymbol{t}_i \right)^\top - \eta \,\text{SoftMax}\,(\boldsymbol{t}_{n+1}) \left( \sum_i a_{n,i} \boldsymbol{t}_i \right)^\top \tag{4}$$

where $\boldsymbol{W}_{OV}^{(s)}$ and $\boldsymbol{W}_{OV}^{(s+1)}$ are the OV-circuit parameters before and after the $s$-th gradient update step and $\eta$ is the learning rate. The positive incremental component in the right-hand side of Equation 4 dictates that, when applied on a attention-weighted linear combination of the context tokens, OV-circuits learn to memorize a linear combination of possible next tokens.

However, in a deep Transformer model with several attention heads, MLP blocks and layer normalization, we can not determine the exact token-token map for the OV-circuits of attention head. Moreover, as Elhage et al. (2021) suggested, multiple attention heads across different layers can construct compositions, where the deeper heads use the output of the shallower heads. Instead, we can assume that, each attention head memorizes to write towards a particular direction in the residual stream when operated upon a sequence of residual stream vectors. One can intuitively call each attention head to be a *mini-LM*. When pretrained using web-sized corpus, these attention heads can memorize undesired token-token associations that hurt the downstream performance, or result in unsafe behavior.

### 3.2 EDITING MODEL BEHAVIOR VIA ATTENTION ROTATION

A natural conclusion from the prior discussion would be that, by suppressing undesired associations for certain attention heads, we can improve task performance. However, multiple token associations are expected to be memorized in each attention head in superposition since the number of attention heads is way smaller than the potential token associations present in the pretraining data — one cannot selectively switch off one certain association. Prior research in mechanistic interpretability has shown that, although we can often localize attention heads responsible for particular task, removing the non-dominant attention heads does not deliver the performance of the full model (Wang et al., 2023; Dutta et al., 2024).

Instead, one can *rotate* the output of the attention heads in order to maximize its alignment with rows of $\boldsymbol{W}_U$ corresponding to certain tokens while near-orthogonalizing with certain undesired tokens. This way, the model behaviour can be edited without destroying the superposed associations. Defining the complete space of $d \times d$ rotation matrices and optimizing them can become computationally challenging. Instead, we utilize the fact that any $d \times d$ orthonormal matrix is similar to a block-diagonal matrix $\boldsymbol{R}_\Theta$, where $\Theta = \{\theta_1, \cdots, \theta_{d/2}\} \subset [0, 2\pi)^{\frac{d}{2}}$, defined as:

$$
\boldsymbol{R}_\Theta^d = \begin{pmatrix}
\cos\theta_1 & -\sin\theta_1 & 0 & 0 & \cdots & 0 & 0 \\
\sin\theta_1 & \cos\theta_1 & 0 & 0 & \cdots & 0 & 0 \\
0 & 0 & \cos\theta_2 & -\sin\theta_2 & \cdots & 0 & 0 \\
0 & 0 & \sin\theta_2 & \cos\theta_2 & \cdots & 0 & 0 \\
\vdots & \vdots & \vdots & \vdots & \ddots & \vdots & \vdots \\
0 & 0 & 0 & 0 & \cdots & \cos\theta_{d/2} & -\sin\theta_{d/2} \\
0 & 0 & 0 & 0 & \cdots & \sin\theta_{d/2} & \cos\theta_{d/2}
\end{pmatrix}
\tag{5}
$$

Given the multi-head attention with $H$ heads at layer $l \in [L]$, where $L$ is the total number of layers in the Transformer, defined as:

$$
\text{Attn}_l(\boldsymbol{x}_n^{(l)}|[\boldsymbol{x}_1^{(l)}, \cdots, \boldsymbol{x}_n^{(l)}]) = \boldsymbol{W}_O \mathop{\Big\|}_{h=1}^{H} \sum_i a_{n,i}^{(h,l)} \boldsymbol{W}_V^{(h,l)} \boldsymbol{x}_i^{(l)}
$$

where $\|$ is the concatenation operator, $a_{n,i}^{(h,l)}$ and $\boldsymbol{W}_V^{(h,l)}$ denote the attention probability between source and destination residual streams at layer $l$ $\boldsymbol{x}_i^{(l)}$ and $\boldsymbol{x}_n^{(l)}$ and the value projection matrix corresponding to the attention head with index $h \in [H]$ at layer $l$, we define the rotated attention as:

$$
\text{RotAttn}_l(\boldsymbol{x}_n^{(l)}|[\boldsymbol{x}_1^{(l)}, \cdots, \boldsymbol{x}_n^{(l)}]) = \boldsymbol{W}_O \boldsymbol{R}_{\Theta_l}^d \mathop{\Big\|}_{h=1}^{H} \sum_i a_{n,i}^{(h)} \boldsymbol{W}_V^{(h)} \boldsymbol{x}_i^{(l)}
\tag{6}
$$

Note that the block-diagonal definition of $\boldsymbol{R}_\Theta^d$ in Equation 5 implies that applying $\boldsymbol{R}_\Theta^d$ on the concatenated head outputs is equivalent to applying $H$-distinct $\boldsymbol{R}_\Theta^{d/H}$ on each of the head outputs.

Without prior knowledge of which attention heads are responsible for memorizing undesired token associations, we need to apply the intervention defined in Equation 6 on a set of attention blocks at layers $l \in \hat{\mathbb{L}}$ (see Section 4 for the choice of the set $\hat{\mathbb{L}}$). Then, the intervened forward pass is denoted as:

$$
\hat{\boldsymbol{t}}_{n+1} = \mathcal{M}_{\text{Rotated}}\left(\{\boldsymbol{t}_1, \cdots, \boldsymbol{t}_n\}|\Theta_{\text{Original}}, \Theta_{\text{Rotation}}\{\Theta_l|l \in \hat{\mathbb{L}}\}\right)
\tag{7}
$$

where $\Theta_{\text{Original}}$ is the set of pretrained model parameters and $\Theta_{\text{Rotation}}$ are the parameters of rotations.

### 3.3 OPTIMIZATION OF ROTATION PARAMETERS

With the rotational interventions defined, all that we are left with is to optimize the rotational parameters. Let $\mathcal{D} := \{\boldsymbol{T}_j, \boldsymbol{Y}_j|j \in [D]\}$ be a set of $D$ supervised examples for a given task, with $\boldsymbol{T}_j$, $\boldsymbol{Y}_j$ referring to the sequence of tokens corresponding to the input and gold output, respectively. If $\boldsymbol{Y}_j = \{\boldsymbol{y}_j\}$ is a single label token, the cost function to optimize becomes straightforward:

$$
\max_{\Theta_{\text{Rotation}}} \sum_j p\left(\mathcal{M}_{\text{Rotated}}\left(\boldsymbol{T}_j|\Theta_{\text{Original}}, \Theta_{\text{Rotation}}\{\Theta_l|l \in \hat{\mathbb{L}}\}\right) = \boldsymbol{y}_j\right)
\tag{8}
$$

In a few-shot setup, the objective function is modified to:

$$\max_{\Theta_{\text{Rotation}}} \sum_{j} p \left( \mathcal{M}_{\text{Rotated}} \left( \overset{M}{\underset{m=1}{\big\|}} [\boldsymbol{T}_m, \boldsymbol{y}_m] \,\big\|\, \boldsymbol{T}_j | \Theta_{\text{Original}}, \Theta_{\text{Rotation}} \{\Theta_l | l \in \hat{\mathbb{L}}\} \right) = \boldsymbol{y}_j \right) \quad (9)$$

In the case of NLG tasks, maximizing the aggregate probability of all the generated tokens can be a solution. However, the goal of our proposed rewiring method is to minimize undesired behaviors. When a model demonstrates such behaviors, depending upon the task, not all tokens equally correspond to the behavior under inspection. The pretrained model is trained using teacher-forcing and is generally able to generate grammatically correct responses. Hence, trying to align the model generation to a single reference response does not make much sense. Instead, we opt for a surrogate scoring function $s : \{\boldsymbol{Y}_j\} \to \mathbb{R}$ that scores the "desirability" of a generated response. We let the model with rotation intervention to generate a complete response given an input, compute the score for the generated response, and seek to minimize the aggregate score across $\mathcal{D}$:

$$\max_{\Theta_{\text{Rotation}}} \sum_{j} s \left( \underset{k}{\big\|} \arg\max \left( \mathcal{M}_{\text{Rotated}} \left( [\boldsymbol{T}_j \,\big\|\, \boldsymbol{Y}_{:k-1}] | \Theta_{\text{Original}}, \Theta_{\text{Rotation}} \{\Theta_l | l \in \hat{\mathbb{L}}\} \right) \right) \right) \quad (10)$$

where $\boldsymbol{Y}_{:k-1}$ denotes the token sequence generated till the $(k-1)$-th decoding step.

We implement Bayesian optimization (Snoek et al., 2012) to solve the optimization problems in Equations 8, 9 or 10 depending upon the task. However, standard Gaussian Process with Matern kernel fails to scale to high dimension input space (Li et al., 2024b). Instead, Infinite-width Bayesian Neiral Networks (I-BNN), proposed by Lee et al. (2017), has shown to scale effectively with high-dimensional parameter space[2]. Furthermore, I-BNN covariance function is not based on Euclidean distance, allowing Gaussian Process to represent non-stationary functions. This is advantageous as effects of rotations may not have similar behaviour throughout the entire configuration space.

## 4 EXPERIMENT SETUP

**Training setting.** Dutta et al. (2024) previously found that token associations corresponding to pretrained knowledge primarily resides in the initial half of the model. Since the rotational intervention designed in Equations 6 and 7 are primarily targeted towards undesired token associations acquired through pretraining, we restrict $\mathbb{L}$ to the initial half only. Therefore, the total number of parameters to optimise becomes $\frac{dL}{4}$. Since we want to optimise the rotation matrix for a particular task, only a small subset of training samples is required, i.e, $6 \le D_{training} \le 20$.

**Models.** Four different instruction-tuned models with varying size are used for all experiments: Qwen2-1.5B-Instruct Yang et al. (2024), Phi-3-mini-4k-instruct Abdin et al. (2024) (2.8 billion parameter), Mistral-7B-Instruct-v0.1 Jiang et al. (2023), and Meta-Llama-3-8B-Instruct Dubey et al. (2024); we refer to these models as `Qwen2-1.5B`, `Phi-3-mini`, `Mistral-7B`, and `Llama-3-8B`, respectively.

**Tasks.** We experiment with five different classification (i.e., single token generation) tasks and two NLG tasks. Classification tasks used are as follows: (1) **AG News:** Classify the corpus of news article into four different categories – World, Sports, Business, Science/Technology (Zhang et al., 2015); (2) **Entailed Polarity:** Test the ability of the model to detect entailed polarity from implicative verbs (Srivastava et al., 2022); (3) **Navigate:** Given a series of navigation instructions, determine whether one would end up back at the starting point (Srivastava et al., 2022); (4) **Color:** Identify the color specified by the given RGB, HEX, HSL, or HCL encoding Srivastava et al. (2022); and (5) **Winowhy:** Evaluate the reasoning in answering Winograd Schema Challenge questions. Of these five tasks, the last four are from BIG-bench collection (BIG-bench authors, 2023). The generation tasks used include (1) **Imdb Positive Review** Maas et al. (2011): Optimise model to produce positive IMDB movie reviews, and (2) **Detoxify** Gehman et al. (2020): Tune the model to generate detoxified text. Further details and examples of tasks are available in Appendix A.1

---

[2]Here the term "high dimension" is relatively used. Our method seeks to optimize only the rotation configurations that scales as $\mathcal{O}(Ld)$, which is substantially low-dimensional if compared to the parameter space of the LM itself.

| Method | AG News | Entailed polarity | Navigate | Color | Winowhy | Avg. |
|---|---|---|---|---|---|---|
| Qwen2-1.5B | | | | | | |
| Base | 0.691 | **1.000** | 0.173 | 0.155 | 0.389 | 0.4816 |
| Eigen Pruning | 0.720 | 0.919 | 0.290 | 0.175 | 0.415 | 0.5038 |
| Rescaling | **0.796** | 0.719 | 0.214 | 0.155 | 0.458 | 0.4684 |
| TaRot | 0.778 | 0.980 | **0.515** | **0.199** | **0.547** | **0.6038** |
| Phi-3-mini | | | | | | |
| Base | 0.729 | **1.000** | 0.470 | 0.253 | 0.588 | 0.6080 |
| Eigen Pruning | 0.519 | 0.878 | 0.392 | 0.270 | 0.099 | 0.4316 |
| Rescaling | 0.739 | 0.921 | 0.273 | **0.295** | **0.629** | 0.5714 |
| TaRot | **0.740** | **1.000** | **0.491** | 0.289 | 0.600 | **0.6240** |
| Mistral-7B | | | | | | |
| Base | 0.653 | 0.762 | 0.140 | 0.431 | 0.618 | 0.5208 |
| Rescaling | 0.437 | **0.896** | **0.550** | 0.219 | 0.683 | 0.5570 |
| TaRot | **0.721** | 0.823 | 0.216 | **0.470** | **0.767** | **0.5994** |
| Llama-3-8B | | | | | | |
| Base | 0.662 | 0.980 | 0.155 | 0.236 | 0.568 | 0.5202 |
| Rescaling | 0.636 | 0.544 | **0.550** | 0.209 | 0.255 | 0.4388 |
| TaRot | **0.718** | **1.000** | 0.464 | **0.459** | **0.701** | **0.6684** |

Table 1: **Overall performance in zero-shot regime.** Performance of methods with different LLMs in terms of F1 scores are presented across different tasks and on average. **Bold-faced** and underlined numbers denote the best and second-best methods. For `Mistral-7B` and `Llama-3-8B`, Eigen Pruning resulted in OOM.

**Baysian optimization.** We use I-BNN with 12 hidden layers, and LogExpectedImprovement as the acquisition function. We use a mixture of $M$-shots generation to avoid biasing the intervention, with $M$ chosen randomly from 0 to 6. Each task was optimized for 150 iterations.

**Baselines.** We compare `TaRot` with three different baselines: (1) *Base model* denotes the pre-trained LLM (zero-shot or few-shot) without any interventions. (2) *Eigen Pruning* (Vergara-Browne et al., 2024) removes singular values from weight matrices in an LLM to improve its performance in a particular task. To have a fair comparison, we also use a maximum of 20 prompts in its training phase. (3) *Rescaling* ablates attention heads by scaling their output in the unit interval instead of rotating their outputs; we use the same optimization technique to figure out the optimal scaling configuration.

**Evaluation metrics.** For NLG tasks, Imdb and Detoxify, two different types of reward models are used. For Imdb positive review tasks, a sentiment analysis reward model, `lvwerra/distilbert-imdb`[3] is used. `Roberta-hate-speech-dynabench-r4-target`[4] is used for detoxification. To calculate the fluency of the generated text, GPT4 Achiam et al. (2023) is used as an oracle to assign a value between 1 and 5, 1 being the least and 5 being the highest. The average of fluency rating is taken to report the number. Further details about the prompts are presented in Appendix A.2

## 5 RESULTS

Tables 1 and 2 summarize the overall performance of different methods across different classification tasks in zero- and 6-shot regimes, respectively. Note that Eigen Pruning is used for comparison in zero-shot only, following their original design. In Table 3, we summarize the results for NLG tasks.

**Consistent improvement with `TaRot`.** Across all different LLMs of varying parameter sizes, `TaRot` demonstrates consistent performance as either the best or second-ranked method across all tasks. Subsequently, we can see the considerable improvement achieved across task-wise average F1 scores: 25.37%, 2.63%, 15.09%, and 28.49% relative improvements compared to the base version of `Qwen2-1.5B`, `Phi-3-mini`, `Mistral-7B`, and `Llama-3-8B`, respectively, in the zero-shot regime (see Table 1). Only in the case of Entailed polarity task using `Qwen2-1.5B`, `TaRot` comes short of improving upon the original model itself (although it scores 0.98 F1 compared to the

---
[3]https://huggingface.co/lvwerra/distilbert-imdb
[4]https://huggingface.co/facebook/roberta-hate-speech-dynabench-r4-target

| Method | AG News | Entailed polarity | Navigate | Color | Winowhy | Avg. |
|---|---|---|---|---|---|---|
| | | | Qwen2-1.5B | | | |
| Base | 0.680 | **0.902** | 0.173 | 0.145 | 0.393 | 0.459 |
| Rescaling | 0.662 | 0.765 | 0.314 | **0.201** | **0.576** | 0.504 |
| TaRot | **0.695** | **0.902** | **0.494** | 0.176 | 0.544 | **0.562** |
| | | | Phi-3-mini | | | |
| Base | 0.745 | 0.974 | 0.440 | 0.372 | 0.604 | 0.627 |
| Rescaling | 0.732 | 0.980 | 0.196 | 0.223 | 0.562 | 0.539 |
| TaRot | **0.764** | **0.991** | **0.494** | **0.402** | **0.647** | **0.660** |
| | | | Mistral-7B | | | |
| Base | 0.691 | 0.921 | **0.236** | 0.346 | **0.790** | 0.597 |
| Rescaling | **0.746** | 0.698 | 0.196 | 0.094 | 0.580 | 0.463 |
| TaRot | 0.684 | **0.960** | 0.196 | **0.464** | **0.790** | **0.619** |
| | | | Llama-3-8B | | | |
| Base | 0.524 | 0.950 | 0.645 | 0.486 | 0.651 | 0.651 |
| Rescaling | 0.444 | 0.702 | 0.196 | 0.093 | 0.577 | 0.402 |
| TaRot | **0.638** | **1.000** | **0.727** | **0.560** | **0.761** | **0.737** |

Table 2: **Overall performance in few-shot regime.** Performance of methods with different LLMs in terms of F1 scores are presented across different tasks (and on average). **Bold-faced** and underlined numbers denote the best and second-ranked methods, respectively.

perfect prediction by the original model). With baseline methods like Eigen Pruning or Rescaling, lack of consistency is a major drawback; while they can improve upon the base model in some cases, drastic deterioration is frequent. Furthermore, there is no task-wise or model-wise pattern of such improvements or failures. For example, Eigen Pruning improves upon Qwen2-1.5B on all tasks except Entailed polarity, but fails drastically with Phi-3-mini on all tasks except color.

**In-context examples vs. TaRot.** Unlike Eigen Pruning (or even, traditional fine-tuning), TaRot is optimized with a mixture of M-shot inference to avoid zero-shot bias. Consequently, we can observe the improvement over the base model achieved via TaRot while provided with in-context examples, except with Mistral-7B on AG News and Navigate (c.f. Table 2). Moreover, in a number of cases, zero-shot TaRot performs comparable to or even better than standard ICL with the original model (e.g., with Llama-3-8B on AG News, Entailed polarity, and Winowhy, with Qwen2-1.5B across all tasks, etc.). The effects of providing ICL examples to the base model or the intervened version with TaRot are not the same across tasks or across models. However, if ICL examples improve the base model, then they improve the TaRot-optimized version as well. A contradictory trend is observable across different models: performance of Qwen2-1.5B and Llama-3-8B (base as well as TaRot-optimized) improve in few-shot regime on the BIG Bench tasks (except Entailed polarity) but deteriorates on AG News, while Phi-3-mini and Mistral-7B show the opposite behavior.

| Method | Imdb | | Detoxify | |
|---|---|---|---|---|
| | Reward | Fluency | Reward | Fluency |
| | Qwen2-1.5B | | | |
| Base | -0.8079 | —- | 4.5025 | —- |
| Rescaling | **0.7245** | 1.255 | **2.2949** | 1.265 |
| TaRot | -0.2511 | **2.24** | 4.0130 | **4.56** |
| | Mistral-7B | | | |
| Base | -0.0561 | —- | 4.3106 | —- |
| Rescaling | **0.1998** | 2.12 | **3.1893** | 4.12 |
| TaRot | 0.1647 | **2.5** | 4.0109 | **4.306** |
| | Llama-3-8B | | | |
| Base | -0.3118 | —- | 4.0533 | —- |
| Rescaling | **0.2800** | 2.56 | **3.1893** | 4.76 |
| TaRot | 0.0015 | 2.387 | 3.9012 | 4.24 |

Table 3: **Performance comparison on NLG tasks.** In IMDB, more positive reward is better; in case of toxicity, smaller reward value is better.

**Importance of rotation over rescaling attention heads.** Comparing TaRot against the rotation-free intervention via Rescaling reveals useful insights regarding the effects of mechanistic intervention. As already mentioned, Rescaling is generally very brittle and there is no predictable pattern in this brittleness. For example, with Mistral-7B in zero-shot Entailed polarity prediction, Rescaling can outperform both base model and TaRot by a large margin (see Table 1) but significantly deteriorates the performance of the rest of the models; moreover, this improvement with Mistral-7B does not scale in the few-shot regime on the same task (see Table 2). Similar patterns are observed with other model-task pairs as well. There are two intertwined factors at play here. First, as explained in Section 3.2, the token associations memorized in the attention heads

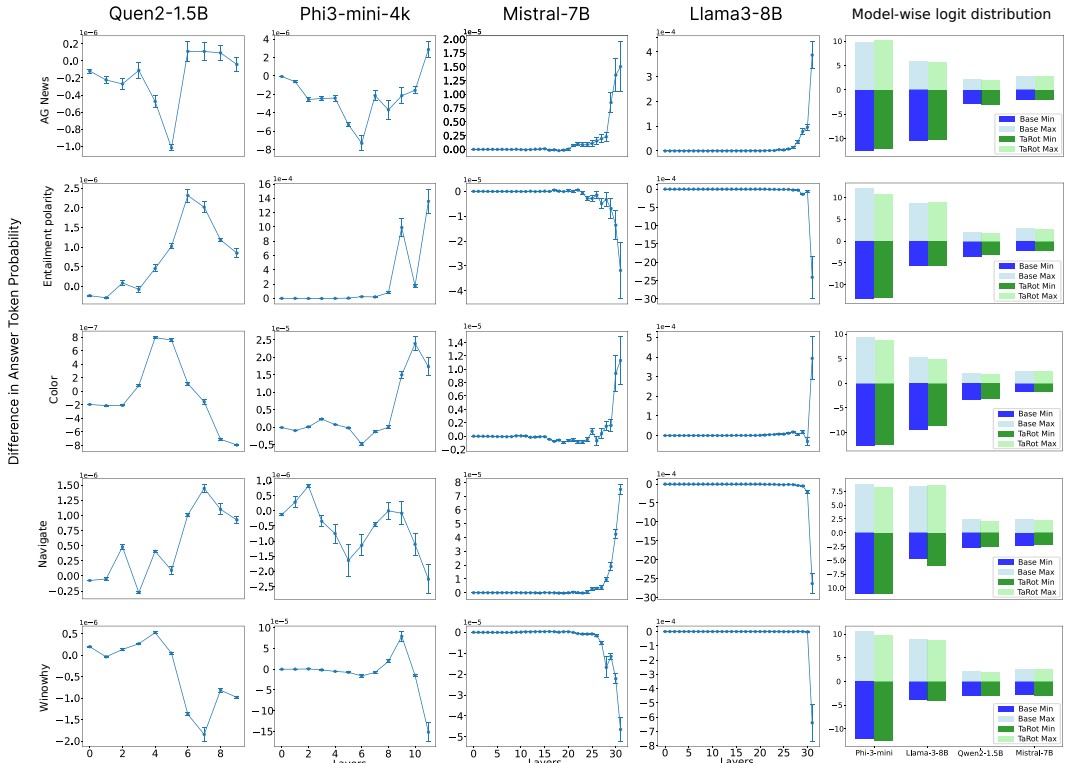

Figure 2: **Change in answer token probability and logit distribution via `TaRot`.** For each model and each task, we plot the difference in the probability of the correct answer token between `TaRot`-intervened and original forward pass at each layer (layer-wise logits are calculated via logit attribution of post-LayerNorm residual stream). Additionally, we plot the mean distribution of the maximum and minimum logit values for each model.

are embedded in a superposed state; directly scaling or ablating them can result in unpredictable behaviors. Second, the possibly large fluctuations introduced by Rescaling render the optimization much harder. Given that the number of parameters to optimize is much smaller in the Rescaling technique compared to `TaRot` (the former needs $H$ parameters per layer, compared to $\frac{d}{2}$ in the latter), the hardness of optimization is in turn primarily dictated by the polysemantic nature of the OV-circuits of the attention heads. For certain tasks in certain setups, downscaling all the token associations for certain heads improves performance — possibly due to the non-interacting nature of those associations with respect to the task. However, this can vary across models and tasks in an unpredictable manner. Instead, the rotational alignment in `TaRot` provides a more fine-grained control over the intervention; subsequently, it behaves in a robust manner. However, we observe an interesting pattern in case of NLG tasks (see Table 3). In terms of reward value, Rescaling seems to perform better than `TaRot`, and both interventions perform better than the original model. However, `TaRot` delivers more fluent response in terms of evaluation by GPT-4. Since Rescaling edits more drastically compared to `TaRot`, higher improvement in terms of task-specific reward is expected. But it costs the model with fluency as it loses on the syntactic nuances, possibly due to tampered syntactic associations. This points out the need for more robust, multi-dimensional evaluation when generation-targeted interventions are concerned.

## 6 ANALYSIS OF ACTION

Towards understanding the nuances of `TaRot`'s action on the neural representation, we start with investigating the probability of the answer token at different layers of the forward pass. Specifically, we adopt *logit attribution* (nostalgebraist, 2020): for a given layer $l$ with output residual stream corresponding to the last token, $\boldsymbol{x}_n^{(l+1)}$, we compute the intermediate probability of the answer token as: $p = \text{SoftMax}\left(\boldsymbol{W}_U \boldsymbol{x}_n^{(l+1)}\right)_{\text{answer}}$. In Figure 2, we plot $p_{\texttt{TaRot}} - p_{\text{Base}}$ for each model across all

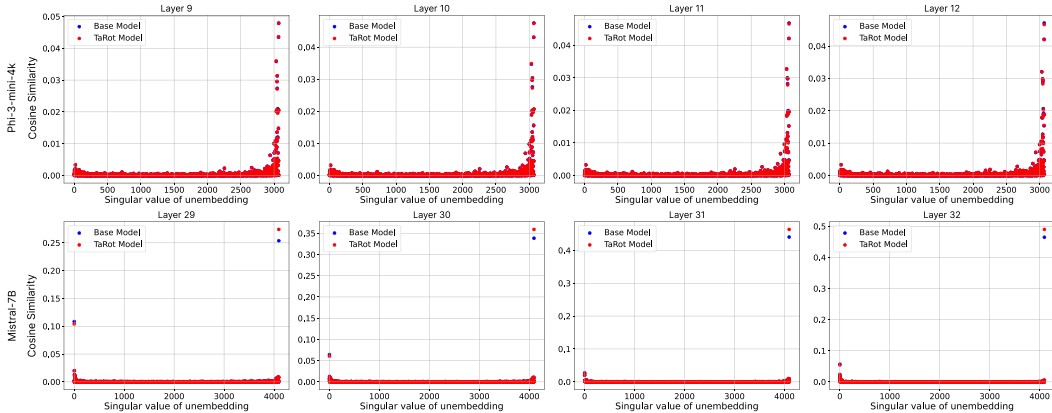

Figure 3: **Impact of `TaRot` on residual subspace.** We plot cosine similarities between the residual stream vectors corresponding to the last token and basis vectors corresponding to the singular values of unembedding (decreasing from left to right) for WinoWhy task (see Appendix A.3 for the rest of the tasks). There is a strong bias to the near-zero singular values, denoting that rotation orthogonalizes certain directions of residual stream.

the layers on different tasks. The overall change in answer token probability remains marginal ($< 10^{-4}$) across all the instances, signifying a key aspect of `TaRot`: it does not substantially improve the desired behavior, rather it minimizes the undesired token associations. However, with `Qwen2-1.5B` and `Phi-3-mini`, there are fluctuations right from the beginning. In case of larger models like `Mistral-7B` and `Llama-3-8B`, probability difference appears only at the very end. Note that negative (or positive) difference in answer token probability does not essentially mean one method is better than the other. Additionally, we plot the distribution of maximum and minimum logit values for each model. Again, there is no significant change in the logit distribution as well, denoting that `TaRot` does not introduce temperature-increment in the logits.

Following Stolfo et al. (2024), we further investigate the impact of `TaRot` on the unembedding subspace[5]. We perform singular value decomposition of $\boldsymbol{W}_U$ into $\boldsymbol{U\Sigma V}^\top$. We then compute the cosine similarity between the residual stream vectors corresponding to different layers and the row vectors of $\boldsymbol{V}^\top$, and plot it alongside the corresponding singular values (see Figure 3). A strong bias is observed where the `TaRot` intervened residual stream aligns more to the smaller singular values of unembedding, thereby decreasing their impact. In `Mistral-7B`, the effect is more skewed compared to `Phi-3-mini`. This observation provides a definitive characterization of `TaRot`'s action on the different subspaces of the residual stream.

## 7 CONCLUSION

In this work, we proposed `TaRot`, a novel, gradient-free, mechanistic intervention method for editing language models. `TaRot` builds on observations from implicit gradient descent bias of causal attention and applies parametrized rotation on the attention output to minimize the effects of undesired memorizations, doing away with effort-intensive localization steps and task-specificity of prior intervention techniques. Using Bayesian optimization of the rotational parameters, `TaRot` renders as data-efficient as in-context learning; yet, across a variety of tasks and language models of different sizes and families, robust improvement is observed. We further analyzed the impact of `TaRot` and demonstrated the key mechanism of action. In a nutshell, `TaRot` can pave the path for general-purpose model editing methods in the future beyond supervised fine-tuning.

**Limitations and ethical considerations.** `TaRot` is designed to perform when the model has a generalization ability that is suppressed by noisy memorization. In that sense, it is limited by the boundaries of pretraining and cannot be used for domain adaptation. Fundamentally, it is not applicable to proprietary models. Finally, similar to any intervention technique, `TaRot` can be used in reverse to bypass alignment tuning and reinforce undesired behaviors.

---

[5]Note that we did not find any unembedding null space like GPT-2 as reported by Stolfo et al. (2024).

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

# A APPENDIX

## A.1 TASK DETAILS

We experimented with five different classification (i.e., single token generation) tasks and two NLG tasks. Below are the details of the tasks with their prompt templates used:

**AG News:** The goal of the task is to categories new articles into one of the four predefined categories.

- World – News about global events, international politics, and worldwide issues.
- Sports – News related to sporting events, athletes, competitions, and sports industry developments.
- Business - News focusing on the economy, financial markets, companies, and business trends.
- Science & Technology – News about technological advancements, scientific discoveries, and research.

**System prompt used for AG News task:** *You are a news classification model. Your task is to classify news articles into one of the following four categories: World, Sports, Business, or Science. You should respond with only the category name and no other characters.*

**Entailed Polarity:** The Entailed Polarity task is a yes/no question-answering task Srivastava et al. (2022). Given a fact and a question, the goal is to determine whether the fact entails a yes or no answer to the question. The task tests the model's ability to infer whether the factual statement logically supports the answer in terms of polarity (positive or negative). Example:

- Fact: "Ed remembered to go."
- Question: "Did Ed go?"
- Answer: "Yes"

**System prompt used for Entailed Polarity task:** *Follow the instructions below and answer with Yes / No.*

**Navigate:** The objective is to follow a set of directional or spatial instructions and determine if, after following those steps, the entity returns to the starting point. The answer is either True or False, depending on whether the instructions guide the entity back to where they started. Example:

- Instruction: "If you follow these instructions, do you return to the starting point?"
- Steps: "Always face forward.", "Take 7 steps left.", "Take 2 steps backward.", "Take 7 steps backward.", "Take 7 steps backward.", "Take 3 steps forward."
- Question: "Do you return to the starting point?"
- Answer: False

**System prompt used for the task:** *Answer the following question and output only True/False.*

**Color:**   This task includes 3,000 random colors written in four common color spaces (RGB, RGB Hex, HSL, and HCL) that we use to probe LLM's knowledge about color encodings. For example, given the prompt *hsl(30.16, 89.56%, 45.91%)*, we expect the model to answer "orange".

**System prompt used for color task:** *Choose the correct color from the options and output the color only.*

**Winowhy:**   This task Srivastava et al. (2022) requires models to identify the correct reasons behind the answers to the Winograd Schema ChallengesZhang et al. (2020).

This task is based on the original Winograd Schema Challenge (WSC) dataset and 4095 WinoWhy reasons (15 for each WSC question) that could justify the pronoun coreference choices in WSC. The model is presented with a passage that contains a pronoun and an explanation of which word or entity the pronoun refers to. The model's job is to assess whether the explanation given is correct or incorrect based on the context of the passage.

- Text: "Fred is the only man alive who still remembers my father as an infant. When Fred first saw my father, he was twelve years old. The 'he' refers to Fred because, in his own words, he is 'a very odd man'."
- Question: "The above reasoning is:"
- Answer: "Incorrect".

**System prompt used for Winowhy task:** *Follow the instructions and output Correct/Incorrect.*

**Imdb:**   Tune model to generate positive movie reviews using a BERT Kenton & Toutanova (2019) sentiment classifier as a reward function. The reward model evaluates the sentiment of the generated reviews, and the goal is to maximize the likelihood of generating reviews classified as positive.

- Dataset Used: imdb Maas et al. (2011)
- Reward Model: lvwerra/distilbert-imdb, a fine-tuned version of distilbert-base-uncased Sanh (2019) on the imdb dataset.

**Detoxify:**   Involves reducing the toxicity of language model outputs. The toxicity evaluation is done using a classifier, such as facebook/roberta-hate-speech-dynabench-r4-target, which distinguishes between "neutral" and "toxic" text. The classifier provides feedback (reward or penalty) based on the toxicity of the model's output, guiding the model to produce less toxic text. The dataset used is allenai/real-toxicity-prompts Gehman et al. (2020).

## A.2   FLUENCY

To evaluate the fluency of a given text, the following prompt was used with GPT4 Achiam et al. (2023): **System prompt used:** *Please rate the fluency of the following text on a scale of 1 to 5, where 1 is least fluent and 5 is most fluent:* text. *Provide only the number.*

where text is the output from the model.

## A.3   COSINE SIMLIARITY

Figures 5, 6, 7, 4 show the impact of `TaRot` on residual subspace for AG News, Color, Entailed Polarity and Navigate tasks, respectively.

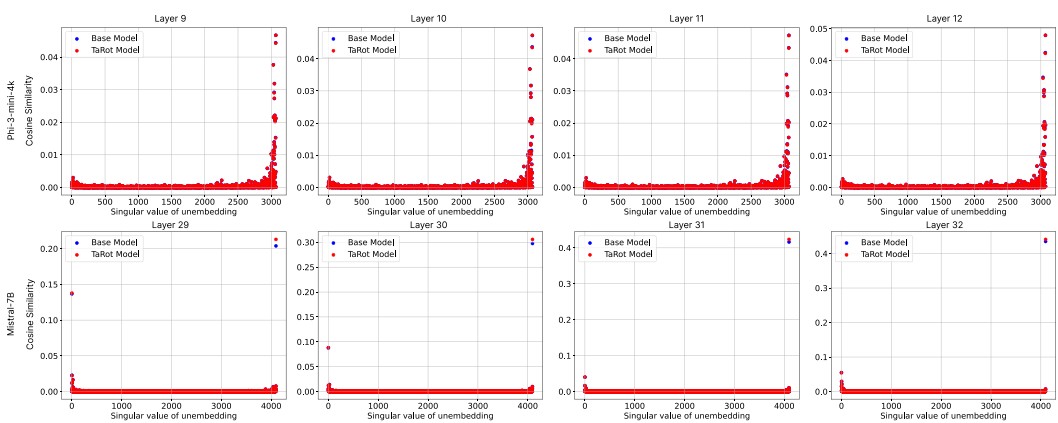

Figure 4: Impact of `TaRot` on residual subspace for Navigate Task

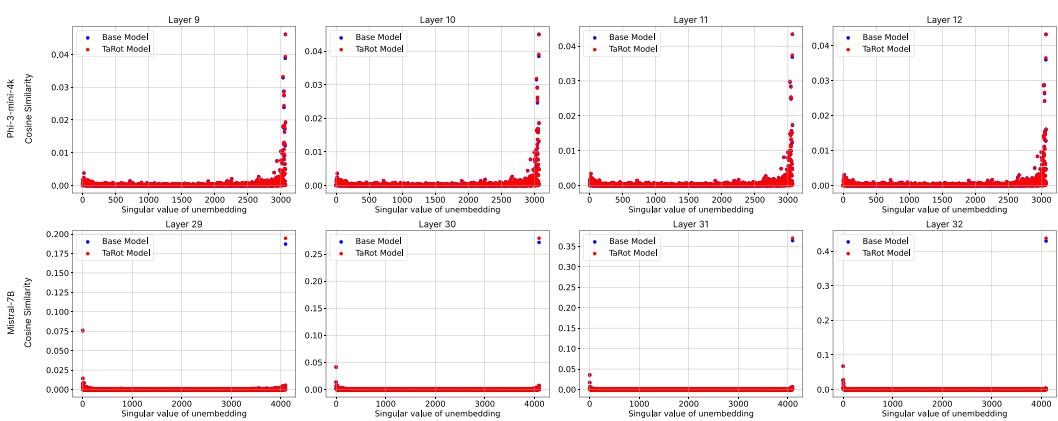

Figure 5: Impact of `TaRot` on residual subspace for AG News task.

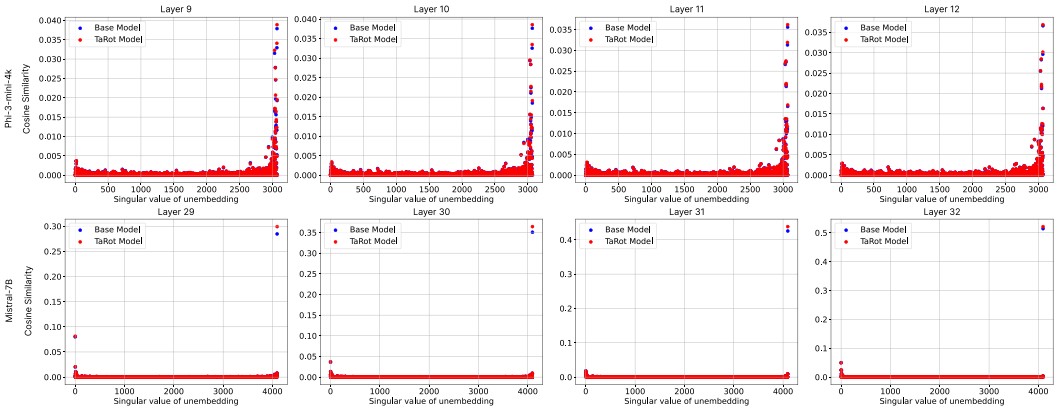

Figure 6: Impact of `TaRot` on residual subspace for Color task.

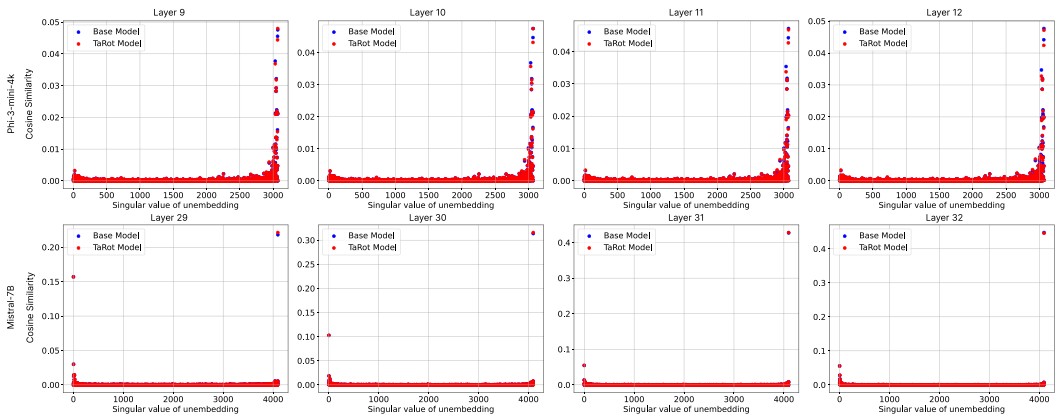

Figure 7: Impact of `TaRot` on residual subspace for Entailed Polarity task.

