# OpenReview forum: "Mechanistic Behavior Editing of Language Models"
_ICLR.cc/2025/Conference — Submitted to ICLR 2025_

### Official Review · Reviewer_aqdz · 2024-10-27

**Soundness:** 3
**Presentation:** 3
**Contribution:** 3
**Rating:** 6
**Confidence:** 3

**Summary:**

This paper proposes TaRot, a method for editing language models. It uses learnable rotation matrices with Bayesian optimization on few shot examples. TaRot is data efficient and demonstrate improvements upon traditional zero-shot and few-shot learning on several classification and generation tasks.

**Strengths:**

- TaRot is an innovative method that is theoretically grounded.
- Strong empirical results demonstrating the benefit of the proposed approach.
- The paper is well written.

**Weaknesses:**

Strong results are observed on simple classification tasks but on generation (positive reviews, detoxification) tasks the results are mixed.

**Questions:**

It would be great to have more analysis/experiments on how the number of available labeled examples and base model size change the behavior of the proposed methods.

---

> ### Author Response · Authors · 2024-11-22
>
> >It would be great to have more analysis/experiments on how the number of available labeled examples and base model size change the behavior of the proposed methods.
>
> To test TaRoT performance on how many training examples should be used, we use varying training size from 2 to 50 shown below. We trained the Qwen 2 1.5B model on navigate task and we see that a very low number of training samples and even a high number of sizes hampers the performance. Therefore we choose a size of size around 20 to achieve the best performance.
> Increasing the size of model’s beyond 8B was not possible due to lack of computational resources. Therefore analysis could only be done using Smaller models.
>
> | Training Data Size | Baseline Zero Shot | TaRoT Zero Shot | Baseline Few Shot | TaRoT Few Shot |
> | ------------------ | ------------------ | --------------- | ----------------- | -------------- |
> | 2                  | 0.17               | 0.17            | 0.17              | 0.21           |
> | 6                  | 0.17               | 0.17            | 0.17              | 0.17           |
> | 10                 | 0.17               | 0.16            | 0.17              | **0.59**           |
> | 20                 | 0.17               | **0.52**            | 0.17              | 0.49           |
> | 30                 | 0.17               | 0.17            | 0.17              | 0.31           |
> | 50                 | 0.17               | 0.17            | 0.17              | 0.17           |
>
> To show superior performance of TaRoT on we ran SFT finetuning using LoRa and ran baseline: RED (https://arxiv.org/pdf/2402.15179). Below tables highlight the performance of SFT compared to RED. To have a direct comparison to our model we kept the number of training samples to be equal as TaRoT samples. Moreover training scripts were only compatible for Llama family models therefore we could run it for only Llama.
>
> | Model                          | Color | Ag_news | Winowhy | Entailed Polarity | Navigate |
> | ------------------------------ | ----- | ------- | ------- | ----------------- | -------- |
> | Llama-3-8B (RED)               | 0.21  | 0.69    | **0.98**    | 0.96              | 0.24     |
> | Llama-3-8B (SFT)               | 0.20  | 0.64    | 0.91    | 0.97              | 0.16     |
> | Llama-3-8B (TaRoT) - Zero Shot | 0.46  | **0.72**    | 0.70    | **1.00**              | 0.46     |
> | Llama-3-8B (TaRoT) - Few Shot  | **0.56**  | 0.64    | 0.76    | **1.00**              | **0.73**     |
> |                                |       |         |         |                   |          |
> | Qwen-2 (SFT)                   | 0.12  | 0.53    | **0.55**    | 0.95              | 0.17     |
> | Qwen-2 (TaRoT) -zero Shot      | **0.20**  | **0.78**    | 0.55    | **0.98**              | **0.52**     |
> | Qwen-2 (TaRoT) -Few Shot       | 0.18  | 0.70    | 0.54    | 0.90              | 0.49     |
>
> We conducted additional experiments on Gemma 9B model and showed its superior performance on TaRoT. We have a limited computational budget to experiment with either TaRot or any other editing/finetuning methods on models larger than this.
>
> | Dataset           | ag_news | winowhy | entailed_polarity | navigate |
> | ----------------- | ------- | ------- | ----------------- | -------- |
> | Base (Zero Shot)  | 0.71    | 0.52    | 1.00              | 0.42     |
> | TaRoT (Zero Shot) | **0.71**    | **0.59**    | **1.00**              | **0.53**     |
> | Base (Few Shot)   | 0.70    | **0.70**    | 1.00              | 0.62     |
> | TaRoT (Few Shot)  | **0.71**    | 0.68    | **1.00**              | **0.64**     |

---

> > ### Author Response · Authors · 2024-11-22
> >
> > >Strong results are observed on simple classification tasks but on generation (positive reviews, detoxification) tasks the results are mixed.
> >
> > Table 3 shows the performance of TaRoT on NLG Tasks. TaRoT performs better than Base models however show inferior performance when compared to rescaling. Upon careful examination of generated output we saw inferior fluency of generated text. Therefore we used GPT 4 to evaluate and score the fluency of text. We see higher fluency scores of TaRoT even though recycling toxicity is lower, pointing towards better generation.
> >
> > | Method (Qwen 2 1.5B)  | Reward | Fleuncy |
> > | --------------------- | ------ | ------- |
> > | Base                  | 4.5025 | 3.62    |
> > | Rescaling             | 2.2949 | 1.265   |
> > | TaRoT                 | 4.013  | 4.56    |
> > |                       |        |         |
> > |                       |        |
> > | **Method (Mistral - 7B)** | **Reward** | **Fleuncy** |
> > | Base                  | 4.3106 | 4       |
> > | Rescaling             | 3.1893 | 4.12    |
> > | TaRoT                 | 4.0109 | 4.306   |
> >
> > Here are some of the examples for detoxification for Mistral -7B model
> >
> > **Text**: The Coffs Coast Advocate reported that the distressed dog drew the family's attention to the little boy's plight.
> >
> > **Generation (rescaling)**: It sounds like you are sharing a news article about a (Fluency: 4)
> >
> > **Generation (TaRoT)**: That's a great example of how animals can be sensitive to our emotions and help us in unexpected ways. (Fluency: 5)
> >
> > **Text:** For years, India has grown up on the hard fact of \"slavery of 200 years\", that refers to the period that the country
> >
> > **Generation (rescaling):** has been undergoing through the 19th and 20th (Fluency: 1)
> >
> > **Generation (TaRoT):** was under British colonial rule. This period is often seen as a time of great suffering and oppression for the (Fluency: 3)

---

> > > ### Author Response · Authors · 2024-11-23
> > >
> > > Dear reviewer aqdz,
> > >
> > > We have addressed all the concerns you raised with additional empirical evidence. We request you to kindly review our responses and let us know if you have further questions.
> > >
> > > We will make our best effort to clarify your doubts and concerns.
> > >
> > > Thanks

---

> > > ### Author Response · Authors · 2024-11-23
> > > **Kindly review our responses**
> > >
> > > Dear reviewer aqdz,
> > >
> > > The discussion period ends very soon. This is our sincere request to read our responses. We tried our best to address all your comments with additional experimental results. Kindly let us know if you have any further questions. If our responses address your concerns, kindly consider reassessing our paper.
> > >
> > > We look forward to your response.
> > >
> > > Thanks

---

> ### Author Response · Authors · 2024-11-24
> **Sincere request to check our responses at least once before the discussion ends tomorrow**
>
> Dear Reviewer aqdz,
>
> The discussion period is ending tomorrow. We haven't gotten any feedback from you about our responses. We tried our best to address all your comments with additional experimental results. We sincerely request that you kindly check our responses and consider reassessing our paper.
>
> We look forward to hearing back from you.
>
> Thanks

---

> > ### Author Response · Authors · 2024-11-25
> >
> > Dear reviewer aqdz,
> >
> > The discussion period ends very soon. This is our sincere request to read our responses. We tried our best to address all your comments with additional experimental results. Kindly let us know if you have any further questions. If our responses address your concerns, kindly consider reassessing our paper.
> >
> > We look forward to your response.
> >
> > Thanks

---

> > > ### Author Response · Authors · 2024-11-26
> > > **Please check our response**
> > >
> > > Dear Reviewer aqdz,
> > >
> > > Since the discussion phase is ending, could you please check our responses and reassess the paper?
> > >
> > > Thanks

---

> > > > ### Author Response · Authors · 2024-11-27
> > > > **Could you please check our responses?**
> > > >
> > > > Dear Reviewer aqdz,
> > > >
> > > > We have not received your feedback on our responses. The discussion deadline is ending soon. It is our sincere request to you to check our responses and consider reassessing our paper kindly. We are ready to address other concerns, if any.
> > > >
> > > > Thanks

---

> > > > > ### Author Response · Authors · 2024-11-28
> > > > > **Subsequent reminder to check our responses**
> > > > >
> > > > > Dear Reviewer aqdz,
> > > > >
> > > > > We have not received any feedback from you on our responses. This is another reminder to check our responses. We sincerely request you check our responses.
> > > > >
> > > > > Thanks

---

> > > > > > ### Author Response · Authors · 2024-11-30
> > > > > >
> > > > > > Dear Reviewer aqdz,
> > > > > >
> > > > > > This is another reminder requesting you to please check our responses.
> > > > > >
> > > > > > Thanks

---

### Official Review · Reviewer_P3bZ · 2024-11-03

**Soundness:** 2
**Presentation:** 3
**Contribution:** 2
**Rating:** 3
**Confidence:** 3

**Summary:**

The paper presents TaRot, a method of task adaption for large language models (LLMs). TaRot applies rotation matrices to attention head outputs, aiming to align token associations for desired outcomes in zero- and few-shot settings.

**Strengths:**

1. This work proposes a new mechanistic intervention method for model adaptation without extensive fine-tuning, highlighting its potential significance.

2. The methodology is well-designed, combining mathematical rigor with empirical validation across diverse tasks.

3. This work demonstrates improvement across multiple LLMs on classification and generation tasks.

4. The paper clearly articulates the TaRot concept and its implications, offering comprehensive descriptions of experiments and results.

**Weaknesses:**

1. There is a lack of comprehensive comparison among different editing methods proposed in prior works. Relying on a single method based on singular values may not provide sufficient insight.

2. Current LLMs often exceed 8 billion parameters such as Llama 3 70B, yet this work does not include experiments on models larger than 8B.

3. The proposed method does not outperform the baseline on the primary metric shown in Table 3.

4. The impact of rotation editing on other previously learned knowledge in LLMs is not evaluated.

**Questions:**

1. What does M represent in Equation 9?

2. Since the rotation matrices apply to any input vectors, how can we ensure that they do not negatively affect performance on tasks that were neither trained nor tested?

3. The relevance of Figure 3 to the claims of this work is unclear. The proposed method and the baseline appear to follow very similar distributions. Could the authors provide further clarification?

---

> ### Author Response · Authors · 2024-11-22
>
> Thank you for your insightful review and positive feedback. Below are our responses to address your concerns and we would be happy to provide any further clarifications as required.
>
>
> >There is a lack of comprehensive comparison among different editing methods proposed in prior works. Relying on a single method based on singular values may not provide sufficient insight.
>
> We compare TaRot with supervised LoRA finetuning and RED (https://arxiv.org/pdf/2402.15179). To have a direct comparison to our model we kept the number of training samples to be equal to TaRoT samples. Moreover, training scripts for RED were only compatible with Llama; therefore, we could run it for only Llama.
>
> | Model                          | Color | Ag_news | Winowhy | Entailed Polarity | Navigate |
> | ------------------------------ | ----- | ------- | ------- | ----------------- | -------- |
> | Llama-3-8B (RED)               | 0.21  | 0.69    | **0.98**    | 0.96              | 0.24     |
> | Llama-3-8B (SFT)               | 0.20  | 0.64    | 0.91    | 0.97              | 0.16     |
> | Llama-3-8B (TaRoT) - Zero Shot | 0.46  | **0.72**    | 0.70    | **1.00**              | 0.46     |
> | Llama-3-8B (TaRoT) - Few Shot  | **0.56**  | 0.64    | 0.76    | **1.00**              | **0.73**     |
> |                                |       |         |         |                   |          |
> | Qwen-2 (SFT)                   | 0.12  | 0.53    | **0.55**    | 0.95              | 0.17     |
> | Qwen-2 (TaRoT) -zero Shot      | **0.20**  | **0.78**    | 0.55    | **0.98**              | **0.52**     |
> | Qwen-2 (TaRoT) -Few Shot       | 0.18  | 0.70    | 0.54    | 0.90              | 0.49     |
>
>
> >Current LLMs often exceed 8 billion parameters such as Llama 3 70B, yet this work does not include experiments on models larger than 8B.
>
> We conducted additional experiments on Gemma 9B model and showed its superior performance on TaRoT. We have a limited computational budget to experiment with either TaRot or any other editing/finetuning methods.
>
> | Dataset           | ag_news | winowhy | entailed_polarity | navigate |
> | ----------------- | ------- | ------- | ----------------- | -------- |
> | Base (Zero Shot)  | 0.71    | 0.52    | 1.00              | 0.42     |
> | TaRoT (Zero Shot) | **0.71**    | **0.59**    | **1.00**              | **0.53**     |
> | Base (Few Shot)   | 0.70    | **0.70**    | 1.00              | 0.62     |
> | TaRoT (Few Shot)  | **0.71**    | 0.68    | **1.00**              | **0.64**     |
>
> >The proposed method does not outperform the baseline on the primary metric shown in Table 3.
>
> Table 3 shows the performance of TaRoT on NLG Tasks. TaRoT performs better than Base models however show inferior performance when compared to rescaling. Upon careful examination of generated output we saw inferior fluency of generated text. Therefore we used GPT 4 to evaluate and score the fluency of text. We see higher fluency scores of TaRoT even though recycling toxicity is lower, pointing towards better generation.
>
> | Method (Qwen 2 1.5B)  | Reward | Fleuncy |
> | --------------------- | ------ | ------- |
> | Base                  | 4.5025 | 3.62    |
> | Rescaling             | 2.2949 | 1.265   |
> | TaRoT                 | 4.013  | 4.56    |
> |                       |        |         |
> |                       |        |
> | **Method (Mistral - 7B)** | **Reward** | **Fleuncy** |
> | Base                  | 4.3106 | 4       |
> | Rescaling             | 3.1893 | 4.12    |
> | TaRoT                 | 4.0109 | 4.306   |
>
> Here are some of the examples for detoxification for Mistral -7B model
>
> **Text**: The Coffs Coast Advocate reported that the distressed dog drew the family's attention to the little boy's plight.
>
> **Generation (rescaling)**: It sounds like you are sharing a news article about a (Fluency: 4)
>
> **Generation (TaRoT)**: That's a great example of how animals can be sensitive to our emotions and help us in unexpected ways. (Fluency: 5)
>
> **Text:** For years, India has grown up on the hard fact of \"slavery of 200 years\", that refers to the period that the country
>
> **Generation (rescaling):** has been undergoing through the 19th and 20th (Fluency: 1)
>
> **Generation (TaRoT):** was under British colonial rule. This period is often seen as a time of great suffering and oppression for the (Fluency: 3)
>
> >What does M represent in Equation 9?
>
> M stands for the LM as a function of input context to output tokens. We apologize for missing to denote explicitly. We will correct it in the revised manuscript.

---

> > ### Author Response · Authors · 2024-11-22
> >
> > >Since the rotation matrices apply to any input vectors, how can we ensure that they do not negatively affect performance on tasks that were neither trained nor tested?
> >
> > To ensure this we tested TaRoT’s effect of rotation on other tasks. Below table shows the performance of SFT and TaRoT model, trained on different tasks and evaluated on color task. Dataset A and B denotes the dataset model is trained on and evaluated on respectively.
> > We see TaRoT consistently performs better than SFT when evaluated on other datasets. Moreover TaRoT performance does not decrease drastically as compared to SFT. For example in Navigate SFT decreases from 33% to 20% whereas taRoT performance remains constant.
> >
> > | model | taskA             | taskB | SFT (20 training Size) | Tarot Zero Shot F1 Score | Tarot Few Shot F1 Score |
> > | ----- | ----------------- | ----- | ---------------------- | ------------------------ | ----------------------- |
> > | phi-3 | color             | color | 0.33                   | 0.34                     | **0.42**                    |
> > | phi-3 | ag_news           | color | 0.25                   | 0.30                     | **0.40**                    |
> > | phi-3 | navigate          | color | 0.21                   | 0.23                     | **0.44**                    |
> > | phi-3 | entailed_polarity | color | 0.24                   | 0.26                     | **0.37**                    |
> > | phi-3 | winowhy           | color | 0.30                   | 0.33                     | **0.38**                    |
> >
> > >The relevance of Figure 3 to the claims of this work is unclear. The proposed method and the baseline appear to follow very similar distributions. Could the authors provide further clarification?
> >
> > We would like to stress on the near-zero singular value (rightmost point on x-axis). In the latter layers, TaRot-equipped representation has a higher average cosine similarity (compared to the original representation) with the U^T vector corresponding to this singular value (see the red scatter point above the blue one). This suggests that TaRot redirects the representation towards the null space.

---

> > > ### Author Response · Authors · 2024-11-23
> > >
> > > Dear reviewer P3bZ,
> > >
> > > We have addressed all the concerns you raised with additional empirical evidence. We request you to kindly review our responses and let us know if you have further questions.
> > >
> > > We will make our best effort to clarify your doubts and concerns.
> > >
> > > Thanks

---

> > > ### Author Response · Authors · 2024-11-23
> > > **Kindly review our responses**
> > >
> > > Dear reviewer P3bZ,
> > >
> > > The discussion period ends very soon. This is our sincere request to read our responses. We tried our best to address all your comments with additional experimental results. Kindly let us know if you have any further questions. If our responses address your concerns, kindly consider reassessing our paper.
> > >
> > > We look forward to your response.
> > >
> > > Thanks

---

> > > ### Author Response · Authors · 2024-11-24
> > > **Please check our responses**
> > >
> > > Dear Reviewer P3bZ,
> > >
> > > The discussion period is ending soon. We haven't gotten any feedback from you about our responses. We tried our best to address all your comments with additional experimental results. We sincerely request that you kindly check our responses and consider reassessing our paper.
> > >
> > > We look forward to hearing back from you.
> > >
> > > Thanks

---

> > > ### Comment · Reviewer_P3bZ · 2024-11-24
> > > **Response to Authors**
> > >
> > > Regarding the first question, SFT serves as a relatively weak baseline in this context because it typically involves tuning more parameters. A better way to show the effect on other tasks would be to compare the proposed method directly to the original model.
> > >
> > > As for the second question, I still couldn't see the trend described by the authors in the figure: the plots for the two cases under study largely overlap. It would be worth considering re-plotting if it could improve clarity.

---

> > > > ### Author Response · Authors · 2024-11-25
> > > >
> > > > **Statistics of training and testing data:**
> > > >
> > > > For all the training of the model’s 20 random question answer pairs were selected. Due to time constraint evaluation was done on 50 randomly selected questions. We made sure that the test set remains constant for the models and experiments conducted.
> > > >
> > > > **Performance on detoxification task:**
> > > >
> > > > For detoxification tasks smaller reward value is better. This is also stated in the manuscript. For Qwen-2 1.5B TaRot has a reward of 4.01, vs  4.50 of the base model. Similarly for Mistral 7B, TaRot performs better than the base model, 4.010 vs  4.310 respectively. Therefore TaRoT performs better than the original model.
> > > >
> > > > **Effect of TaRoT on other tasks:**
> > > >
> > > > We showed the performance of the TaRoT model trained on different tasks and evaluated on colour tasks. To set a better comparison we also evaluated the performance of the original model with zero shot and few shot F1 Score.
> > > >
> > > > **Original model Zero Shot F1 Score: 0.296**
> > > >
> > > > **Original model Few Shot F1 Score: 0.325**
> > > >
> > > > For zero Shot TaRoT we only see a decline for navigate and entailed polarity tasks and that too very minimal. For Few shot TaRoT we don't see any decline in performance of the model.
> > > >
> > > > | model | taskA             | taskB | Tarot Zero Shot F1 Score | Tarot Few Shot F1 Score |
> > > > | ----- | ----------------- | ----- | ------------------------ | ----------------------- |
> > > > | phi-3 | color             | color | 0.34                     | 0.42                    |
> > > > | phi-3 | ag_news           | color | 0.30                     | 0.40                    |
> > > > | phi-3 | navigate          | color | 0.23                     | 0.44                    |
> > > > | phi-3 | entailed_polarity | color | 0.26                     | 0.37                    |
> > > > | phi-3 | winowhy           | color | 0.33                     | 0.38                    |
> > > >
> > > > **Clarification of Trend described in Figure 3:**
> > > >
> > > > Thank you for your suggestion. We will replot the figure with bucket-wise mean values. Furthermore, we will provide results of statistical tests (one-tailed t-tests) performed across layers.
> > > >
> > > > **We hope the above clarification addresses your doubts, we kindly request you to consider revising and improving the score.**

---

> > > > > ### Author Response · Authors · 2024-11-26
> > > > > **Please check our responses**
> > > > >
> > > > > Dear Reviewer P3bZ,
> > > > >
> > > > > Since the discussion phase is ending, could you please check our responses and reassess the paper? We have carefully addressed your recent comments.
> > > > >
> > > > > Thanks

---

> > > > > ### Comment · Reviewer_P3bZ · 2024-11-26
> > > > > **Response to Authors**
> > > > >
> > > > > I would thank the authors for their clarification.
> > > > > I still have the following concerns:
> > > > >
> > > > > **Statistics of training and testing data:** A randomly sampled subset of test examples won't be able to fully evaluate different methods. The author should also mention this specific experimental setup in their main text.
> > > > >
> > > > > **Performance on detoxification task:** Regarding the performance on the detoxification task, the proposed method is still inferior to the baseline method, Rescaling, on the primary metric.
> > > > >
> > > > > **Clarification of Trend described in Figure 3:** My concerns about Figure 3 still remain.

---

> > > > > > ### Author Response · Authors · 2024-11-27
> > > > > >
> > > > > > We thank the reviewer for the response and would like to make the following clarifications:
> > > > > >
> > > > > > **Statistics of training and testing data:**
> > > > > >
> > > > > > We will make sure to include these specific details of the experiment in the main text
> > > > > >
> > > > > > **Performance of detoxification:**
> > > > > >
> > > > > > When comparing TaRot with base model and rescaling it should be evaluated on a combination of both primary metric and fluency. TaRot when compared with base model performs better than base model on both the metrics. Whereas in comparison with rescaling, it shows a decrease in performance on primary metric but maintains a better fluency.

---

> > > > > > > ### Author Response · Authors · 2024-11-27
> > > > > > > **Please check our responses**
> > > > > > >
> > > > > > > Dear Reviewer P3bZ,
> > > > > > >
> > > > > > > Could you please check our responses to your recent comments? We are ready to clarify any further concerns. If our responses address your comments, could you please consider reassessing our paper?
> > > > > > >
> > > > > > > Thanks

---

> > > > > > > > ### Author Response · Authors · 2024-11-28
> > > > > > > > **Subsequent reminder to check our responses**
> > > > > > > >
> > > > > > > > Dear Reviewer P3bZ,
> > > > > > > >
> > > > > > > > This is another reminder to check our responses to your recent comments. Sincerely request you check our responses.
> > > > > > > >
> > > > > > > > Thanks

---

> > > > > > > > > ### Author Response · Authors · 2024-11-30
> > > > > > > > > **Requent to check our recent responses**
> > > > > > > > >
> > > > > > > > > Dear Reviewer P3bZ,
> > > > > > > > >
> > > > > > > > > Could you please check our responses to your recent comments? We are ready to clarify any further concerns. If our responses address your comments, could you please consider reassessing our paper?
> > > > > > > > >
> > > > > > > > > Thanks

---

> > ### Comment · Reviewer_P3bZ · 2024-11-24
> > **Response to Authors**
> >
> > Thanks to the authors for providing clarifications and additional results. Can I know the statistics of the training and testing data used to produce the first table? Also, regarding the third table, it appears that the proposed method doesn't even beat the base model on the primary metrics.

---

> ### Comment · Reviewer_P3bZ · 2024-12-02
> **Response to Authors**
>
> Since my aforementioned concerns regarding the evaluation protocol, the performance of the proposed method, and Figure 3 remain unaddressed, I will maintain my current score. I hope the discussion could give the authors some takeaways for further refining their work.

---

### Official Review · Reviewer_S8sC · 2024-11-03

**Soundness:** 3
**Presentation:** 3
**Contribution:** 2
**Rating:** 6
**Confidence:** 4

**Summary:**

This paper introduces TaRot, a novel method for task adaptation in Large Language Models (LLMs). TaRot intervenes in the neural circuitries using learnable rotation matrices optimized with Bayesian Optimization on few labeled samples. The method aims to improve the generalizability of LLMs by minimizing the effects of undesired memorizations from noisy data. Experiments on various classification and generation tasks demonstrate TaRot's efficacy in enhancing both zero-shot and few-shot performance across different LLMs. The paper also analyzes the changes in neural representation introduced by TaRot, providing insights into its mechanism of action.

**Strengths:**

1. TaRot offers a new perspective on task adaptation by directly editing model behavior through mechanistic interventions, which is a significant departure from traditional SFT.
2. The method is highly data-efficient and designed to work with LLMs of varying sizes, demonstrating its scalability and versatility.
3. The paper provides extensive experimental results, showing consistent improvements in performance across multiple tasks and models.

**Weaknesses:**

1. While the method is innovative, the complexity of understanding and implementing rotation matrices in neural circuits might be a barrier for some researchers and practitioners.
2. TaRot's effectiveness is dependent on the quality of pretraining, which means it may not perform well if the base model has significant limitations. This also limits its applicability in scenarios where models need to be fine-tuned for new domains.
3. The paper could benefit from a more detailed comparison with SFT and other model-editing methods such as [1] and [2].

Please also refer to Questions.

[1] https://arxiv.org/pdf/2202.05262
[2] https://arxiv.org/pdf/2309.05973

**Questions:**

1. Will you open-source the code? With the code, can we easily obtain the optimized model by only providing the original model and data, like SFT?
2. How does TaRot ensure that the rotation matrices do not disrupt other valuable associations?
3. Could you demonstrate how TaRot performs with models of varying pretraining quality, or to clarify the limitations of TaRot in domain adaptation scenarios?
4. How does the selection of 6-20 training samples used for training affect the effectiveness?
5. Regarding data selection.
- What are the criteria that the authors used to select the current datasets?
- Are there any specific tasks or types of data where TaRot has been observed to underperform?
- How does the choice of downstream tasks affect the effectiveness of TaRot in comparison to SFT?
6. The current results seem to only demonstrate the effectiveness of TaRot as an alternative to SFT, so why not compare the results with SFT or few-shot SFT?
7. I believe that  additional experimental results on larger models are needed; otherwise, even SFT won't incur much cost and may be easier to use.
8. Are there any application points for TaRot in instruction-tuning?

Typos: line 083 'pertraining'

---

> ### Author Response · Authors · 2024-11-21
>
> Thank you for your insightful review and positive feedback. Below are our responses to address your concerns and we would be happy to provide any further clarifications as required.
>
> >While the method is innovative, the complexity of understanding and implementing rotation matrices in neural circuits might be a barrier for some researchers and practitioners
>
> Indeed, interpretable insights from neural circuits come at the cost of expert engineering. Most existing representation engineering techniques come with this challenge. To eliminate this manual tuning of each of the rotation parameters we use Bayesian optimization, so that practitioners do not have to manually find relevant features and edit them. Moreover, we also aim to introduce this method as a plug and play python library similar to huggingface PFT, which will significantly lower the barriers for researchers to test and use it.
>
> >TaRot's effectiveness is dependent on the quality of pretraining, which means it may not perform well if the base model has significant limitations. This also limits its applicability in scenarios where models need to be fine-tuned for new domains.
>
> We appreciate the concern of TaRoT performance on model’s with limited pre-trained knowledge. However our method is proposed to eliminate spurious features that are introduced during model training. These features are irrelevant and are even counter-positive to the model's performance. For a model that lacks some existing capability, TaRot cannot introduce it. As a result, it is not applicable for domain adaptation. We have already mentioned this limitation in Section 7
>
> >The paper could benefit from a more detailed comparison with SFT and other model-editing methods
>
> To establish more robustness and superior performance over existing models, we ran SFT finetuning using LoRa and ran baseline: RED ([https://arxiv.org/pdf/2402.15179](https://arxiv.org/pdf/2402.15179)). Below tables highlight the performance of SFT compared to RED. To have a direct comparison to our model we kept the number of training samples to be equal as TaRoT samples. Moreover training scripts were only compatible for Llama family models therefore we could run it for only Llama.
>
> | Model                          | Color | Ag_news | Winowhy | Entailed Polarity | Navigate |
> | ------------------------------ | ----- | ------- | ------- | ----------------- | -------- |
> | Llama-3-8B (RED)               | 0.21  | 0.69    | **0.98**    | 0.96              | 0.24     |
> | Llama-3-8B (SFT)               | 0.20  | 0.64    | 0.91    | 0.97              | 0.16     |
> | Llama-3-8B (TaRoT) - Zero Shot | 0.46  | **0.72**    | 0.70    | **1.00**              | 0.46     |
> | Llama-3-8B (TaRoT) - Few Shot  | **0.56**  | 0.64    | 0.76    | **1.00**              | **0.73**     |
> |                                |       |         |         |                   |          |
> | Qwen-2 (SFT)                   | 0.12  | 0.53    | **0.55**    | 0.95              | 0.17     |
> | Qwen-2 (TaRoT) -zero Shot      | **0.20**  | **0.78**    | 0.55    | **0.98**              | **0.52**     |
> | Qwen-2 (TaRoT) -Few Shot       | 0.18  | 0.70    | 0.54    | 0.90              | 0.49     |
>
> We also evaluate training data trained for SFT with LoRa to reach TaRoT performance. Below table shows the performance of SFT model with varying training dataset size for Phi 3 model and Color Task. For this task and model configuration, TaRoT achieves 34% and 42% F1 score for zero shot and few shot respectively.
>
> | Training Size | SFT  |
> | ------------- | ---- |
> | 10            | 0.23 |
> | 20            | 0.33 |
> | 30            | 0.33 |
> | 40            | 0.32 |
> | 50            | 0.30 |
> | 60            | 0.26 |
> | 70            | 0.33 |
> | 80            | 0.29 |
> | 90            | 0.28 |
> | 100           | 0.41 |
>
> >Will you open-source the code? With the code, can we easily obtain the optimised model by only providing the original model and data, like SFT?
>
> Yes we will be open sourcing the code and checkpoints upon acceptance of the paper. As mentioned above we aim to create this into a python package so that model can be trained like SFT

---

> > ### Author Response · Authors · 2024-11-21
> >
> > >How does TaRot ensure that the rotation matrices do not disrupt other valuable associations?
> >
> > To ensure this we tested TaRoT’s effect of rotation on other tasks. Below table shows the performance of SFT and TaRoT model, trained on different tasks and evaluated on color task. Dataset A and B denotes the dataset model is trained on and evaluated on respectively.
> > We see TaRoT consistently performs better than SFT when evaluated on other datasets. Moreover TaRoT performance does not decrease drastically as compared to SFT. For example in Navigate SFT decreases from 33% to 20% whereas taRoT performance remains constant.
> >
> > | model | taskA             | taskB | SFT (20 training Size) | Tarot Zero Shot F1 Score | Tarot Few Shot F1 Score |
> > | ----- | ----------------- | ----- | ---------------------- | ------------------------ | ----------------------- |
> > | phi-3 | color             | color | 0.33                   | 0.34                     | **0.42**                    |
> > | phi-3 | ag_news           | color | 0.25                   | 0.30                     | **0.40**                    |
> > | phi-3 | navigate          | color | 0.21                   | 0.23                     | **0.44**                    |
> > | phi-3 | entailed_polarity | color | 0.24                   | 0.26                     | **0.37**                    |
> > | phi-3 | winowhy           | color | 0.30                   | 0.33                     | **0.38**                    |
> >
> > ### Data selection
> >
> > The chosen tasks highlight three major directions of model capabilities: 1) AGNews primarily deals with language understanding and world knowledge, 2) BIG Bench tasks seek to investigate LM capabilities beyond task memorization, and 3) Generation tasks to show that the intervention can indeed change the model behavior beyond next token prediction.
> >
> > As the results suggest, in certain tasks and certain models, TaRot indeed underperforms compared to SFT, RED, or the Rescaling baseline. However, on average, TaRot is much more reliable and compute/data-friendly.
> >
> > >I believe that additional experimental results on larger models are needed; otherwise, even SFT won't incur much cost and may be easier to use.
> >
> > We conducted additional experiments on Gemma 9B model and showed its superior performance on TaRoT. We have a limited computational budget to experiment with either TaRot or any other editing/finetuning methods.
> >
> > | Dataset           | ag_news | winowhy | entailed_polarity | navigate |
> > | ----------------- | ------- | ------- | ----------------- | -------- |
> > | Base (Zero Shot)  | 0.71    | 0.52    | 1.00              | 0.42     |
> > | TaRoT (Zero Shot) | **0.71**    | **0.59**    | **1.00**              | **0.53**     |
> > | Base (Few Shot)   | 0.70    | **0.70**    | 1.00              | 0.62     |
> > | TaRoT (Few Shot)  | **0.71**    | 0.68    | **1.00**              | **0.64**     |
> >
> > >Are there any application points for TaRot in instruction-tuning?
> >
> > Unfortunately, we lack the computation budget to experiment with instruction tuning on a reasonable scale. We would like to note that TaRot is not meant for incorporation of new abilities but to eliminate spurious, noisy feature-learning. In that sense, a reasonable method would be to filter an instruction-tuned model using TaRot, not instruction-tuning using TaRot.

---

> > > ### Author Response · Authors · 2024-11-23
> > >
> > > Dear reviewer S8sC,
> > >
> > > We have addressed all the concerns you raised with additional empirical evidence. We request you to kindly review our responses and let us know if you have further questions.
> > >
> > > We will make our best effort to clarify your doubts and concerns.
> > >
> > > Thanks

---

> > > ### Author Response · Authors · 2024-11-23
> > > **Kindly review our responses**
> > >
> > > Dear reviewer S8sC,
> > >
> > > The discussion period ends very soon. This is our sincere request to read our responses. We tried our best to address all your comments with additional experimental results. Kindly let us know if you have any further questions. If our responses address your concerns, kindly consider reassessing our paper.
> > >
> > > We look forward to your response.
> > >
> > > Thanks

---

> > > ### Author Response · Authors · 2024-11-24
> > > **Please check our responses**
> > >
> > > Dear Reviewer S8sC,
> > >
> > > The discussion period is ending soon. We have yet to receive feedback from you regarding our responses. We tried our best to address all your comments with additional experimental results. We sincerely request that you kindly check our responses and consider reassessing our paper.
> > >
> > > We look forward to hearing back from you.
> > >
> > > Thanks

---

> > > > ### Comment · Reviewer_S8sC · 2024-11-25
> > > >
> > > > I would like to thank the authors for the detailed response. However, I still have two concerns:
> > > > - I observed that the improvements of TaRoT over SFT are not consistent. For instance, there's a notable improvement on Navigate, but even a decline on Winowhy compared to SFT. I think it would be beneficial to provide insights into which tasks TaRoT performs better.
> > > > - I believe TaRoT, as an alternative to SFT, needs to have its effectiveness validated on larger models. Otherwise, directly applying SFT+alignment does not pose a significant overhead.
> > > >
> > > > I will keep my score unchanged. Good luck.

---

> > > > > ### Author Response · Authors · 2024-11-25
> > > > >
> > > > > We thank the reviewer for the response and we would like to address the above mentioned concerns. We would like to point out that TaRot need not be considered as an alternative to SFT – it can complement SFT to deal with spurious features learned while pretraining (that SFT cannot necessarily eliminate as prior research show).
> > > > >
> > > > > **Insights in which task TaRoT performs better than SFT**
> > > > >
> > > > > We tested the TaRoT capabilities on a wide range of tasks. The difference in performance boost for TaRoT in different tasks is directly linked to the involvement of pretrained information in the model. TaRoT is proposed to eliminate spurious features that are introduced during model training. These features are irrelevant and are even counter-positive to the model's performance. Winowhy dataset is a reasoning dataset however it relies on a lot of pretrained information and syntax of language. For a model that lacks these existing capabilities, TaRot cannot introduce it, therefore it does not show a huge increase. This is evident by looking at performance of base model (Llama 3 8B) on winowhy task, achieving just 0.568 in zero shot setting. TaRoT is able to eliminate those spurious features and increase the performance to 0.70. On tasks such as navigation, which does not require any prior knowledge, we see a huge improvement in such datasets.
> > > > >
> > > > > Example of Winowhy Task:
> > > > >
> > > > > Question: Please answer the following questions about which words certain pronouns refer to.\n\nJane knocked on the door, and Susan answered it. She invited her to come in. The 'She' refers to susan because she was the first person Susan's family have spoken with in a long time.\nThe above reasoning is
> > > > >
> > > > > Possible answer: [Incorrect | Correct]
> > > > >
> > > > > Example of Navigate Task:
> > > > >
> > > > > Question: If you follow these instructions, do you return to the starting point?\nQ: Take 9 steps. Take 5 steps. Take 2 steps. Take 10 steps. Turn right. Take 10 steps. Take 8 steps.
> > > > >
> > > > > Possible answer: [Yes | No]
> > > > >
> > > > > The above example shows the complexity and prior knowledge of the English language it requires to solve winowhy. We also aim to add these additional details and insights for the rest of the datasets in the updated manuscript.
> > > > >
> > > > > **TaRoT performance on larger models**
> > > > >
> > > > > We strongly believe in TaRoT’s effectiveness and we showed this by running experiments on a wide range of models till size 9B. However, given the time constraint and the unavailability of computational requirements for training a bigger model, we weren't able to test TaRoT on bigger models.
> > > > >
> > > > > **We hope the above clarification addresses your doubts, we kindly request you to consider revising and improving the score.**

---

> > > > > > ### Author Response · Authors · 2024-11-26
> > > > > > **Please check our response**
> > > > > >
> > > > > > Dear Reviewer S8sC,
> > > > > >
> > > > > > Since the discussion phase is ending, could you please check our responses and reassess the paper?
> > > > > >
> > > > > > Thanks

---

> > > > > > > ### Author Response · Authors · 2024-11-27
> > > > > > > **Could you please check our recent responses?**
> > > > > > >
> > > > > > > Dear Reviewer S8sC,
> > > > > > >
> > > > > > > Could you please consider checking our recent responses to your new concerns?
> > > > > > >
> > > > > > > Thanks

---

> > > > > > > > ### Author Response · Authors · 2024-11-28
> > > > > > > > **Subsequent reminder to check our responses**
> > > > > > > >
> > > > > > > > Dear Reviewer S8sC,
> > > > > > > >
> > > > > > > > This is another reminder to check our responses to your recent comments. Sincerely requesting to check our responses.
> > > > > > > >
> > > > > > > > Thanks

---

> > > > > > > > > ### Author Response · Authors · 2024-11-30
> > > > > > > > >
> > > > > > > > > Dear Reviewer S8sC,
> > > > > > > > >
> > > > > > > > > This is another reminder requesting you to please check our responses.
> > > > > > > > >
> > > > > > > > > Thanks

---

> > > > > > > > > > ### Author Response · Authors · 2024-11-30
> > > > > > > > > > **Another reminder to check our responses**
> > > > > > > > > >
> > > > > > > > > > Dear Reviewer S8sC,
> > > > > > > > > >
> > > > > > > > > > Since the discussion phase is ending, could you please check our responses and reassess the paper?
> > > > > > > > > >
> > > > > > > > > > Thanks

---

### Official Review · Reviewer_8G4a · 2024-11-04

**Soundness:** 3
**Presentation:** 2
**Contribution:** 2
**Rating:** 5
**Confidence:** 2

**Summary:**

This paper investigates the challenge of adapting pre-trained Large Language Models (LLMs) to downstream tasks, building on the established understanding that LLMs possess both generalizable and spurious neural circuits, and that fine-tuning often suppresses existing circuits rather than introducing new ones. The core research question posed is: is it possible to directly edit the model behavior via mechanistic interventions in a generalizable manner? To address this, the authors propose TaRot (Task-aware Rotation of token association), a novel method presented as an alternative to instruction tuning, aiming to bypass the need for extensive prompt engineering or computationally expensive full finetuning. TaRot achieves this by applying learned rotation matrices to the output of attention heads in the early layers of the LLM, where pretrained knowledge is believed to reside. These rotations are optimized using Bayesian optimization with a small number of labeled examples. The authors demonstrate that TaRot improves performance on a variety of classification and generation tasks on different LLMs.

**Strengths:**

Steering pretrained LLMs with lightweight and interpretable intervention is of massive importance.

**Weaknesses:**

The central claim of TaRot being a "tuning-free" method achieving effects comparable to finetuning requires further scrutiny.  While TaRot avoids modifying the original LLM weights directly, it introduces and optimizes a set of parameters (rotation matrices) for each task.  This approach parallels other parameter-efficient adaptation methods like adapter-based techniques, which also modify model behavior without full finetuning, albeit in a gradient-based manner.  Thus, the distinction of TaRot as truly "tuning-free" is questionable.

Furthermore, the use of the term "mechanistic intervention" feels somewhat weak in this context. Previous mechanistic intervention approaches typically target specific tasks or capabilities within the model, offering valuable insights and interpretability.  In contrast, TaRot’s aim for general applicability across tasks comes at the cost of reduced interpretability, as it does not focus on isolating or explaining specific neural mechanisms.  A direct comparison with gradient-based methods, in terms of both performance and computational cost, is also lacking.  It's unclear whether TaRot's Bayesian optimization of rotation parameters offers a significant advantage over the efficiency of gradient-based updates in techniques like adapters.

Finally, the paper's emphasis on the "gradient-free" nature of TaRot oversimplifies the optimization process. While the original LLM weights remain fixed, the Bayesian optimization of rotation matrices still involves an iterative search procedure guided by performance feedback.  This indirect use of gradients, through the evaluation of the objective function, means TaRot is not strictly gradient-free. A more nuanced discussion of the computational costs and trade-offs compared to fully gradient-based methods would strengthen the paper.

To summarize, previous "mechanistic invervention" approaches usually have good interpretability since they target specific tasks or capabilities. TaRot is supposed to be general so it does not add much interpretability compared to other methods and it is unknown if it performs better than other gradient-based approaches.

**Questions:**

N/A

---

> ### Author Response · Authors · 2024-11-21
>
> We appreciate your valuable effort to review our paper. Bellow are our responses to the issues raised.
>
> ### TaRoT as truly Tuning-Free
>
> We would like to stress that we do not claim TaRot to be tuning-free, since it is indeed an automatic tuning of token-token association within the model.
>
> ### Mechanistic Intervention
>
> We use the term Mechanistic intervention as opposed to Mechanistic Interpretability. The motivation to use the word ‘Mechanistic’ is to stress the fact that the proposed intervention is motivated by the mechanistic analysis of OV circuits and their role in memorizing token associations. Indeed, the meaning of intervention in this context is different from the intervention techniques that are focused to understand the model behavior, not changing it. That said, we are very much open to your suggestion on a more accurate terminology.
>
> ### Gradient Free nature of TaRoT
>
> We claim it to be gradient-free in the same sense as gradient-based learning typically used by the majority of finetuning methods. With Bayesian optimization, we do not compute the gradient of either the model parameters or the additional parameters of rotation matrices.
>
> ### Discussion on complexity
>
> Thank you for this valuable suggestion. TaRot demonstrates efficiency both in terms of labeled example requirements and the number of parameters tuned compared to LoRA finetuning, all the while demonstrating comparable or better performance across tasks (see below table). We commit to present detailed discussion on the complexity analysis in the revised version.
>
> | Model                          | Color | Ag_news | Winowhy | Entailed Polarity | Navigate |
> | ------------------------------ | ----- | ------- | ------- | ----------------- | -------- |
> | Llama-3-8B (RED)               | 0.21  | 0.69    | **0.98**    | 0.96              | 0.24     |
> | Llama-3-8B (SFT)               | 0.20  | 0.64    | 0.91    | 0.97              | 0.16     |
> | Llama-3-8B (TaRoT) - Zero Shot | 0.46  | **0.72**    | 0.70    | **1.00**              | 0.46     |
> | Llama-3-8B (TaRoT) - Few Shot  | **0.56**  | 0.64    | 0.76    | **1.00**              | **0.73**     |
> |                                |       |         |         |                   |          |
> | Qwen-2 (SFT)                   | 0.12  | 0.53    | **0.55**    | 0.95              | 0.17     |
> | Qwen-2 (TaRoT) -zero Shot      | **0.20**  | **0.78**    | 0.55    | **0.98**              | **0.52**     |
> | Qwen-2 (TaRoT) -Few Shot       | 0.18  | 0.70    | 0.54    | 0.90              | 0.49     |

---

> > ### Author Response · Authors · 2024-11-23
> >
> > Dear reviewer 8G4a,
> >
> > We have addressed all the concerns you raised with additional empirical evidence. We request you to kindly review our responses and let us know if you have further questions.
> >
> > We will make our best effort to clarify your doubts and concerns.
> >
> > Thanks

---

> > ### Author Response · Authors · 2024-11-23
> > **Kindly read our responses**
> >
> > Dear reviewer 8G4a,
> >
> > The discussion period ends very soon. This is our sincere request to read our responses. We tried our best to address all your comments with additional experimental results. Kindly let us know if you have any further questions. If our responses address your concerns, kindly consider reassessing our paper.
> >
> > We look forward to your response.
> >
> > Thanks

---

> > ### Author Response · Authors · 2024-11-24
> > **Please check our responses**
> >
> > Dear Reviewer 8G4a,
> >
> > The discussion period is ending soon. We have yet to receive feedback from you regarding our responses. We tried our best to address all your comments with additional experimental results. We sincerely request that you kindly check our responses and consider reassessing our paper.
> >
> > We look forward to hearing back from you.
> >
> > Thanks

---

> ### Author Response · Authors · 2024-11-25
> **Sincere request to check our responses at least once before the discussion ends tomorrow**
>
> Dear Reviewer 8G4a,
>
> We have been trying to reach out to you with a request to check our responses to your comments. We have meticulously addressed all your concerns. We are sure our responses will address your comments. Since the discussion period will end tomorrow, we would appreciate you checking our responses at least once and reassessing our paper.
>
> Sincerely look forward to hearing from you.
>
> Thanks

---

> ### Comment · Reviewer_8G4a · 2024-11-25
>
> I appreciate the clarifications and new results. I think incorporating some of the clarifications and adjustment of terminology use can help the paper to be more precise. In terms of the LoRA finetuning results, I believe it's going to be a substantial change to the paper content with more extensive discussions regarding setups etc.
>
> I increase my soundness score from 2 to 3 (considering that the authors can polish these contents) but would like to keep the overall assessment unchanged.

---

> > ### Author Response · Authors · 2024-11-25
> > **Seeking clarification**
> >
> > Dear Reviewer 8G4a,
> >
> > While we appreciate your acknowledgement and increasing the soundness score, we are still unsure what additional results/clarification you were expecting to change the overall score of the paper. We clarified the terminologies and added new results as per your suggestions. We are committed to adding all of them to the manuscript. May we know your other concerns which still remain unaddressed?
> >
> > Thank you.

---

> > > ### Author Response · Authors · 2024-11-30
> > > **Gentle reminder on the clarification**
> > >
> > > Dear Reviewer 8G4a,
> > >
> > > While we appreciate your acknowledgement and increasing the soundness score, we are still unsure what additional results/clarification you were expecting to change the overall score of the paper. We made sure to clear up the terminologies and added new results as you suggested. We are committed to adding all of them to the manuscript. Please let us know what other concerns still remain unaddressed.
> > >
> > > Thank you

---

### Author Response · Authors · 2024-12-03

We sincerely thank the reviewers for their thoughtful and positive feedback on our work. As the discussion period is coming to an end, we would like to reiterate key points of the rebuttal and discussion phase:



1.  **Baselines and comparison with SFT:** Through Lora Fine Tuning and additional baselines results we show that TaRot demonstrates efficiency both in terms of labeled example requirements and the number of parameters tuned compared to LoRA finetuning, all the while demonstrating comparable or better performance across tasks

2.  **SFT LoRa with varying training Size:** We also evaluated TaRoT with SFT of varying training dataset size. TaRoT easily beats the SFT model keeping the training size the same. Moreover SFT is only above the TaRoT performance with a comparably high number of training dataset as compared to TaRoT training size.

3.  **TaRoT does not disrupt other valuable associations in the model:** To ensure TaRoT optimized for one task does not lose performance on other tasks. We calculated the performance of TaRoT and SFT trained on various tasks and tested on a different task. We see TaRoT consistently performs better than SFT when evaluated on other datasets. Moreover TaRoT performance does not decrease drastically as compared to SFT. For example in SFT trained on Navigate task, and evaluated on ColoR we see a  decrease from 33% to 20% whereas taRoT performance remains constant.

4.  **TaRoT on larger models:** We conducted additional experiments on Gemma 9B model and showed its superior performance on TaRoT.

5.  **Clarification on TaRoT’s effectiveness on quality of pretrained knowledge:** TaRoT is proposed to eliminate spurious features that are introduced during model training. These features are irrelevant and are even counter-positive to the model's performance. For a model that lacks some existing capability, TaRot cannot introduce it. As a result, it is not applicable for domain adaptation. We have already mentioned this limitation in Section 7,

6.  **Complexity of understanding and implementation of TaRoT:** To eliminate this manual tuning of each of the rotation parameters we use Bayesian optimization, so that practitioners do not have to manually find relevant features and edit them. Moreover, we also aim to introduce this method as a plug and play python library similar to huggingface PFT, which will significantly lower the barriers for researchers to test and use it.

Thanks

---

### Meta-Review · Area_Chair_cqLg · 2024-12-18

**Metareview:**

This paper has received ratings of 3, 5, 6, 6, and the reviewers cited the submission has limited innovation and practical impact. The reviewers also raised the concerns over extensive post-review revisions.

In this paper, the authors propose TaRot (Task-aware Rotation), a method for adapt large language models (LLMs) through learnable rotation matrices optimized using Bayesian optimization. It aims at improving zero- and few-shot performance by addressing spurious features learned during training.

Strength:
- Theoretical novelty and innovation in task adaptation via mechanistic interventions.
- Rigorous methodology and empirical evaluation across multiple tasks.
- Strong results in classification tasks and insights into the mechanism of TaRot.

Area for improvements:
- Inconsistent performance, especially across tasks. As highlighted by multiple reviewers, TaRot shows mixed results, particularly in generation tasks, where it underperforms compared to baselines such as SFT and Rescaling on key metrics.
- Experiments are constrained to models up to 9B parameters, failing to demonstrate the method's effectiveness on state-of-the-art larger-scale LLMs.
- While the paper compares TaRot to SFT and RED, the evaluations lack depth and clarity, especially in terms of computational trade-offs and performance relative to other model editing techniques.
- The proposed approach introduces additional parameters (rotation matrices), calling into question the claim of "tuning-free" adaptation. Furthermore, its impact on unrelated tasks and generalizability remains insufficiently explored.
- Presentation and paper clarity - the reviewers expressed concerns about unclear visualizations (e.g., Figure 3) and the lack of detailed analysis in some areas, such as task-specific performance trends and dataset selection criteria.

**Additional Comments On Reviewer Discussion:**

The authors and reviewers were actively engaged in the discussion phase, leading to a productive rebuttal period. The authors addressed many of these issues during the rebuttal phase by running additional experiments and clarifying terminologies. However, reviewers felt that these efforts did not fully resolve the fundamental concerns, such as inconsistent performance, limited applicability to larger models, and insufficient comparative analysis.

Reviewers acknowledged the authors' clarifications but emphasized the necessity for stronger validation on larger models and clearer insights into task-specific performance. Another reviewer expressed reservations due to unresolved issues with evaluation protocols, inconsistent performance, and the lack of clarity in the trends presented in the results. Lastly, while reviewers appreciated the authors' efforts, they highlighted the mixed outcomes observed in generation tasks as an area needing further improvement.

These concerns underline opportunities for refining the work, and addressing them could significantly strengthen the contribution for future submissions.

---

### Decision · Program_Chairs · 2025-01-22

Reject